# How to Capture Higher-order Correlations? Generalizing Matrix Softmax Attention to Kronecker Computation

**Josh Alman**
Columbia University
New York, NY, USA
josh@cs.columbia.edu

**Zhao Song**
Adobe Research
Seattle, WA, USA
zsong@adobe.com

## Abstract

In the classical transformer attention scheme, we are given three $n \times d$ size matrices $Q, K, V$ (the query, key, and value tokens), and the goal is to compute a new $n \times d$ size matrix $D^{-1} \exp(QK^\top)V$ where $D = \mathrm{diag}(\exp(QK^\top)\mathbf{1}_n)$. Here, $\exp()$ is applied entry-wise and $\mathbf{1}_n$ denotes a length-$n$ vector whose entries are all ones.

Intuitively, attention computation captures pairwise information between words in a sentence, but not higher-order information. Indeed, recent work Sanford et al. (2023) has shown that attention units cannot solve simple problems about detecting triples of connected words.

In this work, we study a generalization of attention which captures triple-wise correlations. The generalization is based on computations involving tensors defined by tuples of words. More formally, given five $n \times d$ size matrices $Q, K_1, K_2, V_1$ and $V_2$ (generalized query, key, and value tokens), our new goal is to compute an $n \times d$ size matrix $D^{-1} \exp(Q(K_1 \oslash K_2)^\top)(V_1 \oslash V_2)$ where $D = \mathrm{diag}(\exp(Q(K_1 \oslash K_2)^\top)\mathbf{1}_{n^2})$ and $K_1 \oslash K_2 \in \mathbb{R}^{n^2 \times d}$ denotes the column-wise Kronecker product of $K_1$ and $K_2$. This generalization is indeed able to solve problems about detecting triple-wise connections that were shown to be impossible for transformers.

The potential downside of this generalization is that it appears as though computations are even more difficult, since the straightforward algorithm requires cubic time in $n$. However, we show that in the bounded-entry setting (which arises in practice, and which is well-studied in both theory and practice), there is actually a near-linear time algorithm. More precisely, we show that bounded entries are both necessary and sufficient for quickly performing generalized computations:

- On the positive side, if all entries of the input matrices are bounded above by $o(\sqrt[3]{\log n})$ then we show how to approximate the "tensor-type" attention matrix in $n^{1+o(1)}$ time.

- On the negative side, we show that if the entries of the input matrices may be as large as $\Omega(\sqrt[3]{\log n})$, then there is no algorithm that runs faster than $n^{3-o(1)}$ (assuming the Strong Exponential Time Hypothesis from fine-grained complexity theory).

We also show that our construction, algorithms, and lower bounds naturally generalize to higher-order tensors and correlations. Interestingly, the higher the order of the tensors, the lower the bound on the entries needs to be for an efficient algorithm. Our results thus yield a natural tradeoff between the boundedness of the entries, and order of the tensor one may use for more expressive, efficient attention computation.

Our constructions make use of a novel connection with a higher-order variant on the kernel density estimation problem. They combine a number of technical tools, including the polynomial method, algebraic geometry codes, and multiparty Merlin-Arthur communication protocols.

# 1 INTRODUCTION

Large language models, such as Transformer Vaswani et al. (2017), BERT Devlin et al. (2018), GPT-1 Radford et al. (2018), GPT-2 Radford et al. (2019), GPT-3 Brown et al. (2020), PaLM Chowdhery et al. (2022), OPT Zhang et al. (2022), GPT-3.5, Bard, GPT-4 OpenAI (2023), Llama Touvron et al. (2023a); Rozière et al. (2023), Llama 2 Touvron et al. (2023b) and its successors, have gained immense importance and found a wide range of applications due to their ability to understand and generate human-like text. These models are trained on massive amounts of text data, enabling them to learn patterns, structures, and nuances of human language. They have applications in many areas, including understanding natural language, content generation, improved human-computer interaction, translation and multilingual communication, and rapid prototyping.

The fundamental computational structure at the core of LLMs is called an attention unit. When a length-$n$ input is given to the attention unit (like a sentence or paragraph of $n$ words), we embed it into three matrices $Q, K, V$ (the query, key, and value token matrices) where each has $n$ rows and $d$ columns. Here $d$ is the feature dimension; one has $d \ll n$ in the long sentence regime. Mathematically, the attention unit computes $D^{-1} \exp(QK^\top)V$, where $D = \mathrm{diag}(\exp(QK^\top)\mathbf{1}_n)$ is a diagonal matrix, $\mathbf{1}_n$ denotes the length-$n$ vector with all entries equal to 1, and $\exp$ is applied entry-wise.

Intuitively, the attention unit is finding pairwise correlations between tokens in the input since it computes inner products between pairs of tokens when computing $QK^\top$. However, if the input data has correlated triples of tokens, it is not clear an attention unit can detect this.

A recent and exciting work Sanford et al. (2023) formalized this intuition. They defined a simple task about learning correlations between triples of words, and showed that attention units are unable to solve it. By contrast, they are able to solve the analogous problem of learning correlations between pairs of words. Toward resolving this, Sanford et al. (2023) proposed a generalization of attention computation:

**Definition 1.1** (Tensor generalization of attention scheme). *Given as input $n \times d$ matrices $Q, K_1, K_2, V_1, V_2$, the goal is to construct another $n \times d$ matrix*

$$D^{-1}A(V_1 \oslash V_2).$$

*Here*

- *$V_1 \oslash V_2 \in \mathbb{R}^{n^2 \times d}$ denotes the column-wise Kronecker product of $V_1$ and $V_2$. Similarly for $K_1 \oslash K_2 \in \mathbb{R}^{n^2 \times d}$ below. (The column-wise Kronecker product of matrices $K_1 \in \mathbb{R}^{n \times d}, K_2 \in \mathbb{R}^{n \times d}$ is a matrix $K := K_1 \oslash K_2 \in \mathbb{R}^{n^2 \times d}$ defined as $K_{i_1+(i_2-1)n,j} := (K_1)_{i_1,j} \cdot (K_2)_{i_2,j}, \quad \forall i_1, i_2 \in [n], j \in [d].$)*

- *$A \in \mathbb{R}^{n \times n^2}$ is the $n \times n^2$ matrix $\exp(Q(K_1 \oslash K_2)^\top/d)$, where $\exp$ is applied entry-wise.*

- *$D \in \mathbb{R}^{n \times n}$ is the $n \times n$ diagonal matrix $\mathrm{diag}(\exp(Q(K_1 \oslash K_2)^\top/d)\mathbf{1}_{n^2})$*

- *$\mathbf{1}_{n^2}$ here denotes a length-$n^2$ vector whose entries are all ones.*

One may naturally view $A$ as an $n \times n \times n$ tensor, which is why we call this a 'tensor generalization'; this view will be important in our proofs below.

In this generalization, entries of the matrix $A$ now correspond to triples of tokens, so one may hope that this generalization can detect triple-wise correlations. And indeed, Sanford et al. (2023) show that this is the case: the tensor generalization gets around their expressivity barrier and is able to detect correlations among triples of tokens.

A fundamental question arises naturally: how quickly can generalized attention computations be performed? The running time of attention computations is critically important, since it forms the time bottleneck of LLM training and inference. By generalizing attention to make it more expressive, have we also made it intractably slow?

To answer this question, we focus on an *approximate* version of the tensor attention computation problem. In practical applications, it is sufficient to approximately perform these computations Child et al. (2019); Kitaev et al. (2020); Wang et al. (2020); Choromanski et al. (2021); Daras et al. (2020);

Katharopoulos et al. (2020); Chen et al. (2021; 2022); Qin et al. (2022); Zandieh et al. (2023); Liu et al. (2023); Zhang et al. (2023); Kacham et al. (2023); Dao et al. (2022); Dao (2023), and this often helps lead to faster algorithms.

**Definition 1.2** (Approximate Tensor Attention Computation ATAttC$(n, d, B, \epsilon_a)$). *Let $\epsilon_a > 0$, $B > 0$ be parameters.Given five matrices $Q, K_1, K_2, V_1, V_2 \in \mathbb{R}^{n \times d}$ that satisfy the following bounded constraints,*

- *$\|Q\|_\infty \leq B$, $\|K_1\|_\infty \leq B$, $\|K_2\|_\infty \leq B$, $\|V_1\|_\infty \leq B$, $\|V_2\|_\infty \leq B$*

*we want to generate a matrix $T \in \mathbb{R}^{n \times d}$ which is able to entry-wisely approximate $D^{-1}AV$, i.e.,*

$$\|T - D^{-1}A(V_1 \oslash V_2)\|_\infty \leq \epsilon_a$$

*Here,*

- *the $\ell_\infty$ norm for a matrix $N \in \mathbb{R}^{n \times d}$ is written as $\|N\|_\infty := \max_{i \in [n], j \in [d]} |N_{i,j}|$, and*

- *the other matrices are defined as in Definition 1.1 above.*

We focus here on the natural setting with $d = O(\log n)$ (so that we are modeling long sequences) and $\epsilon_a = 1/\operatorname{poly}(n)$ (so that one can combine the errors from attention computations over an entire network).

In the case of (non-tensor) attention, the computational complexity of exact and approximate attention computation is very well-understood. Keles et al. (2023) showed that the trivial $O(n^2)$ time algorithm is essentially optimal for exact computation, assuming the Strong Exponential Time Hypothesis (SETH). SETH Impagliazzo & Paturi (2001) is a popular conjecture from fine-grained complexity which posits that one cannot substantially improve our current best algorithms for $k$-SAT; see the survey Williams (2018) for more details. Since $k$-SAT algorithms are very well-studied, it is not commonly believed that major improvements are possible, and so much of fine-grained complexity theory is based on this assumption.

Alman & Song (2023) studied the approximate (non-tensor) attention problem and showed that its complexity depends on the magnitude of the entries of the matrices $Q, K$: If they are smaller than $o(\sqrt{\log n})$, then there is a fast algorithm running in time $n^{1+o(1)}$; this near-linear time algorithm is essentially as fast as one could hope for. On the other hand, if they are at least $\Omega(\sqrt{\log n})$, then there is no algorithm substantially faster than the trivial $O(n^2)$ assuming SETH. This theoretical result mirrors practical observations that bounded entries are essential for fast attention Zafrir et al. (2019); Sun et al. (2019); Katharopoulos et al. (2020); Dettmers et al. (2022b); Xiao et al. (2023); Dettmers et al. (2022a); Perez et al. (2023); Shen et al. (2023).

## 1.1 OUR RESULTS

Our main results tightly resolve the computational complexity of the tensor generalization of attention. Generalizing the situation for (non-tensor) attention, we show that whether or not there is a fast algorithm for AAttC depends on the parameter $B$, the magnitudes of the entries in the query, key, and value matrices.

We first show a lower bound, that when $B \geq \Omega(\sqrt[3]{\log n})$, it is impossible to design a truly subcubic-time algorithm (assuming SETH). Note that the straigtforward algorithm for this problem runs in cubic time, so our result shows that one cannot substantially improve on the straightforward algorithm when the entries have magnitude at least $\Omega(\sqrt[3]{\log n})$.

**Theorem 1.3** (Lower bound, informal version of Theorem B.2). *Assuming SETH, for every $q > 0$, there are constants $C, C_a, C_b > 0$ such that: there is no algorithm running in time $O(n^{3-q})$ for the problem ATAttC$(n, d = C \log n, B = C_b \sqrt[3]{\log n}, \epsilon_a = n^{-C_a})$.*

Our second result is a new algorithm, showing that when $B < o(\sqrt[3]{\log n})$, then there is an almost linear time algorithm for solving the problem.

**Theorem 1.4** (Upper bound, informal version of Theorem E.3). *There is an algorithm (Algorithm 1) that solves ATAttC$(n, d = O(\log n), B = o(\sqrt[3]{\log n}), \epsilon_a = 1/\operatorname{poly}(n))$ in time $n^{1+o(1)}$.*

Our Theorems 1.3 and 1.4 together show that the complexity of ATAttC has a very tight transition at $B = \Theta(\sqrt[3]{\log n})$. When $B < o(\sqrt[3]{\log n})$ is smaller than the threshold, the problem can be solved essentially as quickly as one could hope for, in time $n^{1+o(1)}$. Meanwhile, when $B \geq \Omega(\sqrt[3]{\log n})$ is greater than the threshold, it is *impossible* to achieve a subcubic running time, no matter what algorithmic techniques are used (assuming SETH).

It is exciting that, even for the more expressive tensor generalization of attention, there is a near-linear time algorithm in the bounded entry regime. Interestingly, though, the bound must be smaller than for regular attention: for regular attention to have a near-linear time algorithm, it is necessary and sufficient that $B < \sqrt{\log n}$, whereas for tensor-based attention, we show it is necessary and sufficient that $B < \sqrt[3]{\log n}$.

More generally, for any positive integer $k \geq 2$, we study a higher-order tensor generalization of attention which can detect $k$-wise correlations. (Regular attention corresponds to $k = 2$ and ATAttC corresponds to $k = 3$.) For this problem, we further generalize our results to show that there is a near-linear time algorithm when the entries satisfy $B < \sqrt[k]{\log n}$, and that the trivial $O(n^k)$ time essentially cannot be beaten otherwise. This suggests an intriguing tradeoff between the boundedness of the entries, and the expressiveness of attention we can perform quickly: Given vectors corresponding to tokens for LLM training or inference, we let $B$ be the largest magnitude of an entry, then we select the largest $k$ for which $B < \sqrt[k]{\log n}$, and we can quickly perform $k$-th order attention computations for our tokens, but not higher-order attention.

**Definition 1.5** ($k$-th order generalization of Definition 1.1)**.** *Suppose we are given $n \times d$ matrices $Q, K_1, K_2, \cdots, K_{k-1}$ and $V_1, V_2, \cdots, V_{k-1}$, our target is to construct another $n \times d$ matrix*

$$D^{-1}A(V_1 \oslash V_2 \oslash \cdots \oslash V_{k-1})$$

*Here*

- *$V_1 \oslash V_2 \oslash \cdots \oslash V_{k-1} \in \mathbb{R}^{n^{k-1} \times d}$ is the column-wise tensor product of $V_1, \cdots, V_{k-1}$*

- *$A \in \mathbb{R}^{n \times n^{k-1}}$ is the $n \times n^{k-1}$ size matrix $\exp(Q(K_1 \oslash K_2 \oslash \cdots \oslash K_{k-1})^\top / d)$*

- *$D \in \mathbb{R}^{n \times n}$ is the $n \times n$ diagonal matrix $\mathrm{diag}(\exp(Q(K_1 \oslash K_2 \oslash \cdots \oslash K_{k-1})/d)\mathbf{1}_{n^{k-1}})$*

- *$\mathbf{1}_{n^{k-1}}$ is the length-$n^{k-1}$ vector whose entries are all ones.*

**Roadmap.**

In Section 2, we provide a number of basic notations and definitions. In Section 3, we give a technique overview, summarizing our proofs for both our upper bound result and our lower bound result. In Section 4, we prove the key intermediate results for our lower bound result. Our upper bound result, and the remainder of our lower bound result, are proved in the Appendix.

## 2 PRELIMINARY

**Hadamard Product**

**Definition 2.1** ($\circ$ Hadamard product)**.** *Given $A, B \in \mathbb{R}^{n \times d}$, we use $C := A \circ B$ to denote their entry-wise product, i.e., the matrix $C \in \mathbb{R}^{n \times d}$ given by $C_{i,j} = A_{i,j}B_{i,j}$. We similarly define $\circ$ to denote the entry-wise product of vectors or tensors. This is often called the Hadamard product in the literature.*

**Tensor Operations** Many of our proofs will involve manipulating tensors. Here we introduce three different tensor operations we will frequently use.

**Definition 2.2** ($\odot$ tensor computation)**.** *Given matrices $A \in \mathbb{R}^{n \times d}$, $B \in \mathbb{R}^{n \times d}$, $C \in \mathbb{R}^{n \times d}$, we use $T = A \odot B \odot C$ to denote an $n \times n \times n$ tensor whose entries are given by $T_{i,j,l} := \sum_{a=1}^{d} A_{i,a}B_{j,a}C_{l,a}$, $\forall i \in [n], j \in [n], l \in [n]$.*

We note that a tensor $T$ can be written in the form $A \odot B \odot C$ like this if and only if its tensor rank is at most $d$.

**Definition 2.3** ($\otimes$ Kronecker product). *Given two matrices $K_1 \in \mathbb{R}^{n \times d}$ and $K_2 \in \mathbb{R}^{n \times d}$, we define $K := K_1 \otimes K_2 \in \mathbb{R}^{n^2 \times d^2}$ as follows $K_{i_1+(i_2-1)n, j_1+(j_2-1)d} = (K_1)_{i_1,j_1} \cdot (K_2)_{i_2,j_2}, \quad \forall i_1 \in [n], i_2 \in [n], j_1 \in [d], j_2 \in [d]$.*

In this work, we will primarily use the following column-wise version of the Kronecker product.

**Definition 2.4** ($\oslash$ column-wise Kronecker product). *Given matrices $K_1 \in \mathbb{R}^{n \times d}, K_2 \in \mathbb{R}^{n \times d}$, we define matrix $K := K_1 \oslash K_2 \in \mathbb{R}^{n^2 \times d}$ as follows $K_{i_1+(i_2-1)n, j} := (K_1)_{i_1,j} \cdot (K_2)_{i_2,j}, \quad \forall i_1, i_2 \in [n], j \in [d]$.*

**Matrix Multiplication** Finally, our algorithm will make use of matrix multiplications. For positive integers $n, m, d$, we write $\mathcal{T}_{\mathrm{mat}}(n, d, m)$ to denote the time to multiply a given $n \times d$ matrix $A$ and a $d \times m$ matrix $B$. The straightforward algorithm shows that $\mathcal{T}_{\mathrm{mat}}(n, d, m) = O(ndm)$, and this will suffice for our algorithms here; we will typically apply this when two of $n, m, d$ are very small compared to the third, and in this situation, more clever matrix multiplication algorithm do not yield substantial speedups.

## 3 TECHNIQUE OVERVIEW

Generalizing prior work on the computational complexity of the attention problem to our tensor generalization requires overcoming a number of technical challenges. Here we summarize our approach, with an emphasis on differences with the prior work on (non-tensor) attention that we build on Rubinstein (2018); Katharopoulos et al. (2020); Alman & Song (2023); Sanford et al. (2023).

### 3.1 ALGORITHM

**Tool for the column-wise Kronecker product** We begin by introducing a basic tool for manipulating computations involving the column-wise Kronecker product $\oslash$ (see details in Lemma C.3 below). Define the following matrices.

- Given $A_1 \in \mathbb{R}^{n \times d_1}, A_2 \in \mathbb{R}^{n \times d_1}$, we define $A := (A_1 \oslash A_2) \in \mathbb{R}^{n^2 \times d_1}$.
- Given $B_1 \in \mathbb{R}^{n \times d_2}, B_2 \in \mathbb{R}^{n \times d_2}$, we define $B := (B_1 \oslash B_2) \in \mathbb{R}^{n^2 \times d_2}$.
- We define $C \in \mathbb{R}^{d_1 \times d_2}$ as $C := A^\top B$, and similarly define $C_1 := A_1^\top B_1$ and $C_2 := A_2^\top B_2$.

Then, we prove that we have $C_1 \circ C_2 = C$. Using this identity, $C$ can be computed in time $O(\mathcal{T}_{\mathrm{mat}}(d_1, n, d_2))$ given the matrices $A_1, A_2, B_1, B_2$.

**Approximating $D$** In order to perform generalized attention, we aim to compute the matrix $D = \mathrm{diag}(\exp(Q(K_1 \oslash K_2)^\top / d) \mathbf{1}_{n^2})$. Notice that the intermediate matrix $\exp(Q(K_1 \oslash K_2)^\top / d)$ has $n^3$ entries. We thus cannot compute it in subcubic time. We instead aim to use an *implicit* representation of an *approximation* of this matrix which can be quickly manipulated.

Toward this goal, we find appropriate matrices $U_1, U_2, U_3$ (which we discuss in more detail shortly) and formulate $\widetilde{D} = \mathrm{diag}(U_1(U_2 \oslash U_3)^\top \mathbf{1}_{n^2})$ such that $\widetilde{D} \approx D$. Given the matrices $U_1, U_2, U_3$, and using the above tool for $\oslash$, we can compute $\widetilde{D}$ quickly in $O(nd)$ time.

**Approximating $A$** We can similarly approximate the attention matrix $A = \exp(Q(K_1 \oslash K_2)^\top / d)$ via $\widetilde{A} = U_1(U_2 \oslash U_3)^\top$ such that $\widetilde{A} \approx A$ (in $\ell_\infty$ norm). Again, in contrast to $\widetilde{D}$, we cannot compute the entries of $\widetilde{A}$ since it has $n^3$ entries. We instead directly approximate $A(V_1 \oslash V_2)$ by computing $\widetilde{A}(V_1 \oslash V_2)$. This can again be done in $O(\mathcal{T}_{\mathrm{mat}}(d, n, d)) = n^{1+o(1)}$ time by using the above tool.

**Finding approximating matrices $U_1, U_2, U_3$** Thus, it remains to find matrices $U_1, U_2, U_3$ which appropriately approximate $D$ and $A$ as above. We show how to efficiently find such matrices as long as the inputs $Q, K_1, K_1, V_1, V_2$ have bounded entries. The key idea is to use the *polynomial method*, a key tool from prior work Aggarwal & Alman (2022); Alman & Song (2023) which allows one to find low-rank representations of matrices.

The method generally says that if $M$ is a low-rank matrix, and $p$ is a low-degree polynomial, then $p(M)$ (where $p$ is applied entry-wise) also has relatively low rank. Furthermore, its low-rank decomposition can be found efficiently given the decomposition of $M$. By applying this method where $p$ is an appropriate polynomial approximation of the $\exp$ function (see Aggarwal & Alman (2022)), we get a low-rank approximation of $\exp(M)$.

This polynomial method approach was also taken in the prior work on (non-tensor) attention Alman & Song (2023). Here we generalize it, showing that the same line of attack can be applied to low-rank tensors. Viewing $A$ interchangeably as both an $n \times n \times n$ tensor and an $n \times n^2$ matrix allows us to take advantage of this low-rank tensor approximation as well as the aforementioned matrix multiplication algorithms, and $U_1, U_2, U_3$ are ultimately the low-rank approximation expression for this tensor. Notably, as the bound $B$ on the entries increases, the degree of the polynomial to approximate $\exp$ also increases, but the degree needs to be small enough to give a nontrivial algorithm. See details in Lemma E.1.

## 3.2  HARDNESS

**Gap-MaxIP**  Our hardness proof proceeds by introducing and considering a new intermediate problem we call $\mathsf{Gap-MaxIP}$ (Definition 4.6), a promise version of the more common 3-Max IP problem. In this problem, one is given as input $3n$ vectors $a_1, \ldots, a_n, b_1, \ldots, b_n, c_1, \ldots, c_n \in \{0,1\}^d$ as well as a threshold $t$, and the goal is to distinguish between the cases

- $\langle a_i, b_j, c_k \rangle \leq t$ for all $i, j, k \in [n]$, or
- $\langle a_i, b_j, c_k \rangle \geq 2t$ for some $i, j, k \in [n]$.

(If neither is the case, we may give any output.) Here, $\langle a_i, b_j, c_k \rangle$ denotes the 3-way inner product $\sum_{\ell=1}^{d} a_i[\ell] \cdot b_j[\ell] \cdot c_k[\ell]$.

We first prove that $\mathsf{Gap-MaxIP}$ cannot be solved in truly subcubic time assuming SETH. We then show that a truly subcubic time algorithm for our generalized ATAttC (Definition 1.2) problem with large entries would yield one for $\mathsf{Gap-MaxIP}$ as well.

Previous work on (non-tensor) attention Alman & Song (2023) used as its intermediate problem the approximate Hamming Nearest Neighbor problem. However, it is not obvious how to directly generalize this to the tensor setting, since there is no way to define a 'distance' function for triples of vectors which satisfies the needed properties to generalize the original proof. We instead investigate the $\mathsf{Gap-MaxIP}$ problem, which can itself be seen as a generalization of an intermediate step in the proof of hardness for approximate Hamming Nearest Neighbor Rubinstein (2018).

**Hardness of** $\mathsf{Gap-MaxIP}$  Fine-grained complexity results for approximation problems like $\mathsf{Gap-MaxIP}$ have previously been shown using a *distributed probabilistically checkable proof* framework Abboud et al. (2017); Rubinstein (2018), which we also use here.

We begin by generalizing the approach of Rubinstein (2018) using Merlin-Arthur (MA) communication protocols (Babai (1985); Goldwasser & Sipser (1986); Arora & Barak (2009)). We construct a four party communication protocol for the *disjointness* problem: Alice, Bob and Charlie are each given subsets of a universe, and want to determine whether there is an element in all three of their sets. In an MA protocol, Merlin first sends an advice string to the three players to convince them their sets are disjoint. Alice, Bob and Charlie may then flip private random coins and communicate to come to an answer. (See details in Theorem 4.5).

Generalizing known three-party protocols for disjointness Aaronson & Wigderson (2009); Rubinstein (2018), our protocol is algebraic in nature, and critically makes use of algebraic geometry codes from coding theory Shum (2000); Shum et al. (2001).

We then use this protocol to reduce from SAT to $\mathsf{Gap-MaxIP}$. A standard reduction Williams (2005) shows that SAT reduces to the 3OV problem, which is a computational version of the three player disjointness problem. We can convert inputs to this problem into vectors by corresponding entries of the vectors to possible transcripts of the communication protocol. The gap in inner products will arise naturally from the correctness guarantees of the protocol. See reduction details in Theorem 4.7 and its proofs.

**Reducing from** Gap−MaxIP **to** ATAttC    Finally, we reduce the Gap−MaxIP (Definition 4.6) problem to our ATAttC (Definition 1.2) problem. The key idea is that, by defining the matrices $Q, K_1, K_2, V_1, V_2$ of generalized attention in terms of the inputs to Gap−MaxIP, we can make large entries of the attention matrix $A$ correspond to the triples with largest inner product. (See Lemma B.1 below for an illustration.) Some manipulation similar to prior work Alman & Song (2023) allows us to detect large entries from the output of ATATtC. This approach has been used for the fine-grained hardness of many attention and kernel density estimation problems Backurs et al. (2018); Katharopoulos et al. (2020); Alman et al. (2020); Aggarwal & Alman (2022); Alman & Song (2023). See details in Lemma B.1 and its proofs.

## 4    HARDNESS

In this section, we begin the formal proof of our hardness result. We begin by introducing the fine-grained hypotheses we will use.

**Hypothesis 4.1** (Strong Exponential Time Hypothesis (SETH), Impagliazzo & Paturi (2001))**.** *For every $\epsilon > 0$ there exists an integer $k \geq 3$ such that* CNF − SAT *on formulas with clauses size at most $k$ (the so called $k$-SAT problem) and $n$ variables cannot be solved in $O(2^{(1-\epsilon)n})$ time even by a randomized algorithm.*

**Definition 4.2** (3OV)**.** *Given three sets $A, B, C \subset \{0, 1\}^d$ where $|A| = |B| = |C| = n$, the goal is to find a tuple $(i_1, i_2, i_3) \in [n] \times [n] \times [n]$ such that $\langle a_{i_1}, b_{i_2}, c_{i_3} \rangle = 0$.*

**Conjecture 4.3** (Orthogonal Vectors Conjecture (3OVC) Williams (2005); Abboud et al. (2014))**.** *For every $\epsilon > 0$, there is a $c \geq 1$ such that* 3OV *cannot be solved in $n^{3-\epsilon}$ time on instances with $d = c \log n$.*

It is known that SETH implies 3OVC; see, e.g., Williams (2005).

### 4.1    ALGEBRAIC GEOMETRY CODES FROM PREVIOUS WORK

We state a important tool from the field of algebraic geometry codes. For more background on algebraic geometry codes, we refer the reader to Goppa (1981); Tsfasman et al. (1982); Shum (2000); Shum et al. (2001); Sudan (2013).

**Theorem 4.4** (Shum et al. (2001); see also Rubinstein (2018))**.** *There is a constant $q_0 \in \mathbb{N}$ such that, for every prime $q \geq q_0$, there are two systematic code families $\mathcal{C} := \{C_n\}$ and $\mathcal{C}' := \{C'_n\}$ whose codewords are given by functions $w : \mathcal{R}_n \to \mathbb{F}_{q^2}$ for some appropriate subset $\mathcal{R}_n \subset \mathbb{F}_{q^2}^{O(\log n)}$. The codes $\mathcal{C}, \mathcal{C}'$ satisfy four key properties:*

- *Systematicity. There exists a subset $\mathcal{S}_n \subset \mathcal{R}_n$ of cardinality $|\mathcal{S}_n| = \Theta(n)$, such that for any assignment $x : \mathcal{S}_n \to \mathbb{F}_{q^2}$, there exists a codeword $w \in \mathcal{C}$ such that $w|_{\mathcal{S}_n} = x$*

- *3-way Polynomial Closure. $\mathcal{C}$ and $\mathcal{C}'$ are linear codes. For each $w_1, w_2, w_3 \in \mathcal{C}$, there exists $w' \in \mathcal{C}'$ such that for each $i \in \mathcal{R}_n$, $w'(i) = w_1(i) \cdot w_2(i) \cdot w_3(i)$*

- *Efficiency. Both codes can be encoded in $\mathrm{poly}(n)$ time and checked in $\mathrm{poly}(n)$ time.*

- *Parameters. Both codes have relative rate at least $0.01$ and relative distance at least $0.01$.*

### 4.2    A FOUR PARTY MA COMMUNICATION PROTOCOL

Prior work (Rubinstein (2018)) constructed a protocol for three party communication, which includes Merlin, Alice and Bob. Here we modify this protocol for four parties.

**Theorem 4.5.** *For any $T \in [2, m]$. There is a* MA*-communication protocol for Set Disjointness over universe $[m]$. This protocol is computationally efficient.*

*In particular, the details of protocol are*

- *Merlin sends Alice $O(\frac{m \log T}{T})$ bits*

- *Alice, Bob, Charlie toss $O(\log m)$ coins*

- *Charlie sends Alice $O(T \log T)$ bits*

- *Bob sends Alice $O(T \log T)$ bits*

- *Alice returns Accept or Reject*

*If the three sets do not have any element in common, Alice always accepts. Otherwise, she accepts with probability at most $1/2$.*

*Proof.* We assume that $T$ divides $m$, i.e., there is some positive integer $r$ such that $m = Tr$. Otherwise, increase $m$ to the nest multiple of $T$; this at most doubles $m$. We partition the universe into $T$ disjoint sets of size $r$: $[m] = U^1 \cup \cdots \cup U^T$. Let $\alpha, \beta, \gamma \subseteq [m]$ denote the inputs of Alice, Bob, and Charlie. Our goal is to determine whether there is an element in the intersection $\alpha \cap \beta \cap \gamma$.

For each $t \in [T]$, we define the $t$-th parts of the three sets: $\alpha^t := \alpha \cap U^t, \beta^t := \beta \cap U^t, \gamma^t := \gamma \cap U^t$. We will next encode these parts using an algebraic geometry code. Let $q$ be a prime greater than $T$, and let $C$ be an algebraic geometry code over the field $\mathbb{F}_{q^2}$, and let $C'$ be its associated code for the polynomial closure property. Let $\rho_C, \delta_C$ be the rate and distance of the code; recall these are at least a positive constant. Let $n_C = \frac{m}{T \cdot \rho_C} = O(m/T)$ be the length of the codewords of $C$.

For each $t \in [T]$, we write $C(\alpha^t), C(\beta^t), C(\gamma^t)$ to denote the encodings of $\alpha^t, \beta^t$ and $\gamma^t$. Thus, their entry-wise product $\mu^t$ ( i.e., $\mu_i^t := C(\alpha^t)_i \cdot C(\beta^t)_i \cdot C(\gamma^t)_i$ ) is a codeword in the second code $C'$. Furthermore, since $C'$ is a linear code, the entry-wise sum of the $\mu^t$'s ($\mu_i = \sum_{t=1}^T \mu_i^t$) is also a codeword of $C'$.

$C$ is a systematic code, so we may assume that for each $i \in [n/T]$, the entries $C(\alpha^t)_i, C(\beta^t)_i, C(\gamma^t)_i$ are from $\{0, 1\}$ and represent membership in the set. Similarly, $\mu_i^t \in \{0, 1\}$, and the sets are disjoint if and only if $\mu_i^t = 0$ for all $i \in [m/T]$ and $t \in [T]$, or equivalently, $\mu_i = 0$ for all $i \in [m/T]$.

Now the protocol proceeds as follows:

- Step 1. Merlin sends Alice $\widehat{\mu}$, which is supposed to be the encoding of $\mu$

- Step 2. Charlie, Bob and Alice pick a random $i^* \in [n_C]$

- Step 3. Charlie sends Alice $C(\gamma^t)_{i^*}$ for all $t \in [T]$

- Step 4. Bob sends Alice $C(\beta^t)_{i^*}$ for all $t \in [T]$

- Step 5. Alice accepts iff all of the following hold:

  - $\widehat{\mu}$ is a codeword in $C'$ , $\widehat{\mu}_{i^*} = \sum_{t=1}^T C(\alpha^t)_{i^*} \cdot C(\beta^t)_{i^*} \cdot C(\gamma^t)_{i^*}$ and $\widehat{\mu}_i = 0$ for all $i \in [m/T]$

First, we observe that Merlin's message length is $n_c \cdot \log T = O((\log T) \cdot m/T)$ , and both Bob and Charlie's message lengths are $T \cdot O(\log T)$, as desired. To see correctness, note that if Alice ever accepts given Merlin's message $\widehat{\mu}$, then $\widehat{\mu}$ must in particular be a codeword of $C'$. If Alice accepts with probability greater than $1 - \delta_{C'}$ (where $\delta_{C'}$ is a positive constant) then $\widehat{\mu}$ is also equal to the true $\mu$ by definition of $\delta_{C'}$. This means $\mu_i = 0, \forall i \in [m/T]$, so the sets are disjoint.

$\square$

## 4.3 Showing 3-MAX-IP is hard

We define the appropriate gap 3-MAX-IP problem, which we use as our intermediate hard problem.

**Definition 4.6** (Gap approximate maximum inner product search ($\mathsf{Gap-MaxIP}(n, d, t, \epsilon)$))**.** *Suppose the following conditions hold*

- *We use $t > 0$ to represent a threshold parameter.*

- *We use $\epsilon$ to represent an accuracy parameter.*

- *Suppose $n, d$ denote two positive integers.*

- *Given three sets of points, $A = \{a_1, \cdots, a_n\}$, $B = \{b_1, \cdots, b_n\}$, $C = \{c_1, \cdots, c_n\} \subset \{0,1\}^d$*

*For every index $i \in [n]$, we need to distinguish the following two cases*

- *Case 1. There exists a pair $(j_1, j_2) \in [n] \times [n]$ such that $\langle a_i, b_{j_1}, c_{j_2} \rangle \geq t$.*

- *Case 2. For all pairs $(j_1, j_2) \in [n] \times [n]$ we have $\langle a_i, b_{j_1}, c_{j_2} \rangle \leq (1 - \epsilon) \cdot t$.*

Implicit in previous work (Rubinstein (2018)) is a proof that the analogue of $\mathsf{Gap{-}MaxIP}$ with two sets of points is hard. Here we generalize this to three sets.

**Theorem 4.7.** *Unless* SETH *and* OVC *are false, the following holds: for every $\delta > 0$ there are constants $\alpha_1 > \alpha_2 > 0$ such that for integer $n$, solving $\mathsf{Gap{-}MaxIP}(n, d = \alpha_1 \log n, t = \alpha_2 \log n, \epsilon = 1/2)$ requires time $\Omega(n^{3-\delta})$.*

*Proof.* We reduce from 3 OV to $\mathsf{Gap{-}MaxIP}$. Let $\delta_{\mathsf{OV}} = \delta/2$. Our reduction takes as input an instance $(A_{\mathsf{OV}}, B_{\mathsf{OV}}, C_{\mathsf{OV}})$ of orthogonal vectors over $\{0,1\}^m$. These sets have sizes $|A_{\mathsf{OV}}| = |B_{\mathsf{OV}}| = |C_{\mathsf{OV}}| = 2^{m/c}$ for a constant $c$ depending on $\delta_{\mathsf{OV}}$ from Definition 4.2 and Conjecture 4.3, and 3OVC posits there is no algorithm solving this problem in time $O((2^{m/c})^{3-\delta_{\mathsf{ov}}})$.

For a constant $k > 0$ to be determined, pick $\epsilon > 0$ to be a constant such that $\frac{kc \log^2 \log(1/\epsilon)}{\log(1/\epsilon)} < \delta/2$.

We use the protocol of Theorem 4.5, instantiated with parameter $T = T(\epsilon) = O(\frac{\log(1/\epsilon)}{\log \log(1/\epsilon)})$.

Suppose that $T' = 2^{O((\log T) \cdot T)}$ is representing the number of different possible messages sent by Bob and Charlie in the protocol. Let us choose $T$ so that $T' = O(1/\epsilon)$. For each vector $\gamma \in C_{\mathsf{OV}}$, we construct a new vector $\widetilde{c}^\gamma \in \{0,1\}^{(T')^2 \times m}$ by setting $\widetilde{c}^\gamma_{i_B, i_C, j} := 1$ iff Charlie send message $i_C \in [T']$ on input $\gamma'$ and randomness $j \in [m]$. (The value is independent of $i_B$.)

For each vector $\beta \in B_{\mathsf{OV}}$, we construct a new vector $\widetilde{b}^\beta \in \{0,1\}^{(T')^2 \times m}$ by setting $\widetilde{b}^\beta_{i_B, i_C, j} := 1$ iff Bob sends message $i_B \in [T']$ on input $\beta'$ and randomness $j \in [m]$. (The value is independent of $i_C$.)

For each Merlin-message $\mu \in \{0,1\}^{O((\log T) \cdot m/T)}$ and vector $\alpha \in A_{\mathsf{OV}}$, we construct a new vector $\widetilde{a}^{\mu,\alpha} \in \{0,1\}^{(T')^2 \times m}$ as follows: $\widetilde{a}^{\mu,\alpha}_{i_B, i_C, j} := 1$ iff Alice accepts on

- input $\alpha$,

- message $\mu$ from Merlin,

- message $i_B$ from Bob, message $i_C$ from Charlie, and randomness $j$.

Notice also that the inner product of three vectors $\langle \widetilde{a}^{\mu,\alpha}, \widetilde{b}^\beta, \widetilde{c}^\gamma \rangle$ is exactly proportional to the probability that Alice, Bob and Charlie accept on inputs $\alpha, \beta, \gamma$ and message $\mu$ from Merlin.

In particular, if $\alpha$, $\beta$ and $\gamma$ are not orthogonal (i.e., $\langle \alpha, \beta, \gamma \rangle > 0$), then the inner product is at most $\langle \widetilde{a}^{\mu,\alpha}, \widetilde{b}^\beta, \widetilde{c}^\gamma \rangle \leq m/2$. Otherwise, there exists a $\mu$ that Merlin could send to make the players accept, meaning that $\langle \widetilde{a}^{\mu,\alpha}, \widetilde{b}^\beta, \widetilde{c}^\gamma \rangle = m$.

In particular, these can be distinguished by an algorithm for

$$\mathsf{Gap{-}MaxIP}(n = 2^{m/c} \cdot 2^{O(m \log^2 \log 1/\epsilon / \log 1/\epsilon)}, d = 2(T')^2 m, t = m, \epsilon = 1/2),$$

which must therefore be as hard as solving the original instance of 3OV. By 3OVC, this means it requires time

$$(|A_{\mathsf{OV}}| + |B_{\mathsf{OV}}| + |C_{\mathsf{OV}}|)^{3-\delta_{\mathsf{ov}}} = (2^{m/c})^{3-\delta_{\mathsf{ov}}} = n^3 / 2^{m(\delta_{\mathsf{ov}}/c - O(\frac{\log^2 \log(1/\epsilon)}{\log(1/\epsilon)}))} \leq n^{3-\delta}$$

where the last step follows from choosing $k$ large enough in the definition of $\epsilon$.

At the end, we notice that the vectors we construct have dimension $2(T')^2 \cdot m = O(m) = O(\log n)$ as desired. $\qquad\square$

## ACKNOWLEDGEMENTS

The authors would like to thank Yichuan Deng, Yeqi Gao, Junze Yin, Lichen Zhang, Ruizhe Zhang, Tianyi Zhou for helpful discussions of attention literature.

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

APPENDIX

**Roadmap.**

In Section A, we provide the definitions of several notations. In Section C, we provide the running time proofs for our upper bound result. In Section D, we provide the error analysis for our upper bound result. In Section E, we combine everything together, and also present our algorithm. In Section B. we show how to reduce our problem to $\mathsf{Gap-MaxIP}$.

## A PRELIMINARY

We write $\mathbb{R}$ to denote the real numbers, and write $\mathbb{R}^{n \times d}$ to denote the set of $n \times d$ matrices whose entries are real numbers.

For any positive integer $n$, we write $[n]$ to denote $\{1, 2, \cdots, n\}$.

For a matrix $A \in \mathbb{R}^{n \times d}$ and indices $i \in [n]$, $j \in [d]$, we write $A_{i,j}$ to denote the entry of $A$ in the $i$-th row and $j$-th column.

We use $\mathbf{1}_n$ to denote a length-$n$ vector whose entries are all ones.

For a vector $w \in \mathbb{R}^n$, we use $\mathrm{diag}(w) \in \mathbb{R}^{n \times n}$ denote a diagonal matrix with $(\mathrm{diag}(w))_{i,i} = w_i$; all the other (off-diagonal) entries of the matrix are zero.

If $D \in \mathbb{R}^{n \times n}$ is a diagonal matrix, we write $D^{-1} \in \mathbb{R}^{n \times n}$ for its inverse, which is the diagonal matrix whose $i$-th entry on the diagonal is $1/D_{i,i}$, and whose off-diagonal entries are all zero.

For a matrix $A \in \mathbb{R}^{n \times d}$, we use $A^\top \in \mathbb{R}^{d \times n}$ to denote its transpose.

For a vector $x \in \mathbb{R}^n$, we use $\exp(x) \in \mathbb{R}^n$ to denote the entry-wise exponential of $x$, i.e., the length-$n$ vector with $\exp(x)_i = \exp(x_i)$ for all $i \in [n]$.

For a matrix $X \in \mathbb{R}^{n \times n}$, we similarly use $\exp(X) \in \mathbb{R}^{n \times n}$ to denote the matrix with $\exp(X)_{i,j} = \exp(X_{i,j})$.

For any matrix $A \in \mathbb{R}^{n \times d}$, we define $\|A\|_F := (\sum_{i=1}^n \sum_{j=1}^d A_{i,j}^2)^{1/2}$ to be the Frobenius norm of $A$.

For a vector $a, b \in \mathbb{R}^n$, we write $\langle a, b \rangle$ to denote their inner product, $\sum_{i=1}^n a_i b_i$.

For a matrix $A \in \mathbb{R}^{n \times d}$, we write $\|A\|_\infty$ to denote its $\ell_\infty$ norm, i.e., $\|A\|_\infty := \max_{i \in [n], j \in [d]} |A_{i,j}|$.

For a tensor $T \in \mathbb{R}^{n \times n \times n}$, we similarly write $\|T\|_\infty$ to denote the $\ell_\infty$ norm of that tensor $T$, i.e., $\|T\|_\infty := \max_{i \in [n], j \in [n], l \in [n]} |T_{i,j,l}|$.

**Definition A.1** (k-wise inner product). *For $k$ vectors $a_1, \ldots, a_k \in \mathbb{R}^d$, we define*

$$\langle a_1, \ldots, a_k \rangle := \sum_{\ell=1}^d \prod_{i=1}^k a_{i,\ell}.$$

**Definition A.2** (3-MAX-IP). *Given three sets $A, B, C \subseteq \{0, 1\}^d$ of vectors where $|A| = |B| = |C| = n$, the goal is to compute*

$$\max_{a \in A, b \in B, c \in C} \langle a, b, c \rangle$$

## B HARDNESS: FROM MaxIP TO OUR PROBLEM

In Section B.1, we show how to reduce our problem to $\mathsf{Gap-MaxIP}$. In Section B.2, we present our main lower bound (hardness) result.

### B.1 REDUCTION

We now generalize the hardness proof of Alman & Song (2023) to the tensor attention case.

**Lemma B.1.** *For every constant $C_\gamma \in (0, 0.1)$, every $\epsilon > 0$, and every $C > C_0 > 0$, there exist constants $C_a > 0$ and $C_b > 0$ and such that, if $\mathsf{ATAttC}$ (Definition D.1) for parameters $(2n, d = 2C \log n, B = C_b \sqrt[3]{\log n}, \epsilon_a = n^{-C_a})$ can be solved in time $T$, then $\mathsf{Gap-MaxIP}(n, d = C \log n, t = C_0 \log n, \epsilon)$ (Definition 4.6) can be solved in time $O(T + n^{3-C_\gamma})$.*

*Proof.* We give an algorithm for $\mathsf{Gap-MaxIP}(n, d = C \log n, t = C_0 \log n, \epsilon)$ (Definition 4.6). Let $a_1, \cdots, a_n, b_1, \cdots, b_n, c_1, \cdots, c_n \in \{0, 1\}^d$ denote the inputs to this problem. Using them, we will construct appropriate inputs to the $\mathsf{ATAttC}$ problem so that its output will help us to detect triples with large inner product.

Let $\beta > 0$ and $\widetilde{d} \geq d$ be parameters to be determined (in Eq. (5) and Eq. (2) below). Define $\tau > 0$ by

$$\tau := \exp(\beta/2). \tag{1}$$

We pick these parameters so that $\tau$ will be an upper bound on entries of the attention matrix, namely

$$\tau \geq \max_{i \in [n], j_1 \in [n], j_2 \in [n]} \exp(\beta \langle a_i, b_{j_1}, c_{j_2} \rangle / \widetilde{d}).$$

We will use an algorithm for the $\mathsf{ATAttC}(\widetilde{n}, \widetilde{d}, B, \epsilon_a)$ problem with parameters:

$$\widetilde{n} := 2n, \quad \widetilde{d} := 2d, \tag{2}$$

$$B := C_b \sqrt[3]{\log n}, \quad \text{where} \quad C_b := \sqrt{40C/(C_0 \epsilon)}, \tag{3}$$

$$\epsilon_a := n^{-C_a}, \quad \text{where} \quad C_a := 2 + C_b^2 (1 - C_0/C). \tag{4}$$

Furthermore, set

$$\beta := B^3. \tag{5}$$

We define the query and key matrices, $Q \in \mathbb{R}^{\widetilde{n} \times \widetilde{d}}$ and $K_1, K_2 \in \mathbb{R}^{\widetilde{n} \times \widetilde{d}}$ as

$$Q := \sqrt[3]{\beta} \cdot \begin{bmatrix} a_1^\top & \mathbf{1}_d^\top \\ a_2^\top & \mathbf{1}_d^\top \\ \vdots & \vdots \\ a_n^\top & \mathbf{1}_d^\top \\ \mathbf{0}_d^\top & \mathbf{1}_d^\top \\ \mathbf{0}_d^\top & \mathbf{1}_d^\top \\ \vdots & \vdots \\ \mathbf{0}_d^\top & \mathbf{1}_d^\top \end{bmatrix}, \quad K_1 := \sqrt[3]{\beta} \cdot \begin{bmatrix} b_1^\top & \mathbf{0}_d^\top \\ b_2^\top & \mathbf{0}_d^\top \\ \vdots & \vdots \\ b_n^\top & \mathbf{0}_d^\top \\ \mathbf{0}_d^\top & \mathbf{1}_d^\top \\ \mathbf{0}_d^\top & \mathbf{1}_d^\top \\ \vdots & \vdots \\ \mathbf{0}_d^\top & \mathbf{1}_d^\top \end{bmatrix}, \quad \text{and} \quad K_2 := \sqrt[3]{\beta} \cdot \begin{bmatrix} c_1^\top & \mathbf{0}_d^\top \\ c_2^\top & \mathbf{0}_d^\top \\ \vdots & \vdots \\ c_n^\top & \mathbf{0}_d^\top \\ \mathbf{0}_d^\top & \mathbf{1}_d^\top \\ \mathbf{0}_d^\top & \mathbf{1}_d^\top \\ \vdots & \vdots \\ \mathbf{0}_d^\top & \mathbf{1}_d^\top \end{bmatrix}.$$

Since each entry of $Q$ and $K_1, K_2$ is either $\sqrt[3]{\beta}$ or $0$, it follows that

$$\|Q\|_\infty \leq \sqrt[3]{\beta} = B$$

$$\|K_1\|_\infty \leq \sqrt[3]{\beta} = B$$

$$\|K_2\|_\infty \leq \sqrt[3]{\beta} = B$$

$$\|QK^\top/\widetilde{d}\|_\infty \leq \frac{\beta \cdot \widetilde{d}}{\widetilde{d}} = \beta = B^3.$$

In terms of these matrices $Q \in \mathbb{R}^{\widetilde{n} \times \widetilde{d}}$ and $K = K_1 \oslash K_2 \in \mathbb{R}^{\widetilde{n}^2 \times \widetilde{d}}$, the attention matrix

$$A := \exp(QK^\top/\widetilde{d}) \in \mathbb{R}^{\widetilde{n} \times \widetilde{n}^2}$$

is naturally partitioned into eight submatrices

$$A = \begin{bmatrix} A_1 & A_2 & A_3 & A_4 \\ A_5 & A_6 & A_7 & A_8 \end{bmatrix}$$

where each $A_i$ ($\forall i \in [8]$) is a matrix of size $n \times n^2$, defined as follows. For each $j_0 \in [n], j_1 \in [n], j_2 \in [n]$,

- The $(j_0, j_1 + (j_2 - 1)n)$-th entry of $A_1$ is

  - $\exp(\beta(\langle a_{j_0}, b_{j_1}, c_{j_2} \rangle + \langle \mathbf{1}_d, \mathbf{0}_d, \mathbf{0}_d \rangle)/\widetilde{d}) = \exp(\beta \langle a_{j_0}, b_{j_1}, c_{j_2} \rangle/\widetilde{d})$

- The $(j_0, j_1 + (j_2 - 1)n)$-th entry of $A_2$ is

  - $\exp(\beta(\langle a_{j_0}, b_{j_1}, \mathbf{0}_d \rangle + \langle a_{j_0}, \mathbf{0}_d, \mathbf{1}_d \rangle)/\widetilde{d}) = \exp(0) = 1$

- The $(j_0, j_1 + (j_2 - 1)n)$-th entry of $A_3$ is

  - $\exp(\beta(\langle a_{j_0}, \mathbf{0}_d, c_{j_2} \rangle + \langle a_{j_0}, \mathbf{1}_d, \mathbf{0}_d \rangle)/\widetilde{d}) = \exp(0) = 1$

- The $(j_0, j_1 + (j_2 - 1)n)$-th entry of $A_4$ is

  - $\exp(\beta(\langle a_{j_0}, \mathbf{0}_d, \mathbf{0}_d \rangle + \langle \mathbf{1}_d, \mathbf{1}_d, \mathbf{1}_d \rangle)/\widetilde{d}) = \exp(\beta d/\widetilde{d}) = \tau$

- The $(j_0, j_1 + (j_2 - 1)n)$-th entry of $A_5$ is 1

  - $\exp(\beta(\langle \mathbf{0}_d, b_{j_1}, c_{j_2} \rangle + \langle \mathbf{1}_d, \mathbf{0}_d, \mathbf{0}_d \rangle)/\widetilde{d}) = \exp(0) = 1$

- The $(j_0, j_1 + (j_2 - 1)n)$-th entry of $A_6$ is 1

  - $\exp(\beta(\langle \mathbf{0}_d, b_{j_1}, \mathbf{0}_d \rangle + \langle a_{j_0}, \mathbf{0}_d, \mathbf{1}_d \rangle)/\widetilde{d}) = \exp(0) = 1$

- The $(j_0, j_1 + (j_2 - 1)n)$-th entry of $A_7$ is 1

  - $\exp(\beta(\langle \mathbf{0}_d, \mathbf{0}_d, c_{j_2} \rangle + \langle a_{j_0}, \mathbf{1}_d, \mathbf{0}_d \rangle)/\widetilde{d}) = \exp(0) = 1$

- The $(j_0, j_1 + (j_2 - 1)n)$-th entry of $A_8$ is

  - $\exp(\beta(\langle \mathbf{0}_d, \mathbf{0}_d, \mathbf{0}_d \rangle + \langle \mathbf{1}_d, \mathbf{1}_d, \mathbf{1}_d \rangle)/\widetilde{d}) = \exp(\beta d/\widetilde{d}) = \tau$

For each $(i, j_1, j_2) \in [n] \times [n] \times [n]$, we know that

$$
\begin{aligned}
A_{i, j_1 + (j_2 - 1)n} &= \exp(\beta \cdot \langle a_i, b_{j_1}, c_{j_2} \rangle/\widetilde{d}) \\
&\leq \exp(\beta \cdot \|a_i\|_\infty \cdot \|b_{j_1}\|_\infty \cdot \|c_{j_2}\|_\infty \cdot d/\widetilde{d}) \\
&\leq \exp(\beta/2) \\
&= \tau.
\end{aligned}
\tag{6}
$$

Here we used the fact that $d < \widetilde{d}$ (see Eq. (2)), and the last step uses the definition of $\tau$ (see Eq. (1)). We also know that for each $(i, j_1, j_2) \in [n] \times [n] \times [n]$,

$$
A_{i, j_1 + (j_2 - 1)n} \geq 0
\tag{7}
$$

since it is the exponential of an entry of $QK^\top/\widetilde{d}$.

By combining our expression for $A$ with Eq. (6) and Eq. (7), we see that

$$
n^2 \tau \leq (A\mathbf{1}_{\widetilde{n}^2})_i \leq 4n^2 \tau, \quad \forall i \in [\widetilde{n}].
$$

Since $D_{i,i} = (A\mathbf{1}_{\widetilde{n}})_i$, it follows that

$$
n^2 \tau \leq D_{i,i} \leq 4n^2 \tau, \quad \forall i \in [\widetilde{n}].
$$

Choose the vector $v \in \mathbb{R}^{\widetilde{n}^2}$ (recall that $\widetilde{n}^2 = 4n^2$) defined as

$$
v = \begin{bmatrix} \mathbf{1}_{n^2} \\ \mathbf{0}_{n^2} \\ \mathbf{0}_{n^2} \\ \mathbf{0}_{n^2} \end{bmatrix}.
$$

We define $\widetilde{t}$ as

$$\widetilde{t} := \frac{1}{3}\exp(0.25\beta t/d)/(4n^2\tau). \tag{8}$$

We can see that $\widetilde{t} \geq \epsilon_a$ as follows:

$$\begin{aligned}
\widetilde{t} &= \frac{1}{12n^2}\exp(0.25\beta t/d - \beta/2) \\
&= \frac{1}{12n^2}\exp(-0.5\beta + 0.25\beta t/d) \\
&= \frac{1}{12n^2}\exp(-0.5\beta + 0.25\beta C_0/C) \\
&= \frac{1}{12}\exp(-0.5\beta + 0.25\beta C_0/C - 2\log n) \\
&= \frac{1}{12}\exp(-0.5C_b^2\log n + 0.25C_b^2(C_0/C)\log n - 2\log n) \\
&\geq n^{-C_a} \\
&= \epsilon_a.
\end{aligned}$$

Here, the last two steps follow from Eq. (4).

We now use an algorithm for $\mathsf{ATAttC}(\widetilde{n}, \widetilde{d}, B, \epsilon_a)$, with the value matrix $V$ with one row $v$ and the rest $0$. Since $\widetilde{t} \geq \epsilon_a$, the result is a vector $u \in \mathbb{R}^{\widetilde{n}}$ such that, for all $i \in [\widetilde{n}]$,

$$|u_i - (D^{-1}Av)_i| < \widetilde{t}.$$

Recall that for each $i \in [n]$ we need to distinguish between two cases: either there is a pair $(j_1, j_2) \in [n]$ such that $\langle a_i, b_{j_1}, c_{j_2}\rangle \geq t$, or else for all pairs $(j_1, j_2) \in [n]$ the inner product $\langle a_i, b_{j_1}, c_{j_2}\rangle \leq (1 - \epsilon_a)t$. We will distinguish between these cases by checking whether $u_i$ is greater than a threshold value $\widetilde{t}_0 := 2\widetilde{t}$. We next consider the two cases to see why this is.

**Case 1.**

For a given $i \in [n]$, if there are $(j_1, j_2) \in [n] \times [n]$ such that $\langle a_i, b_{j_1}, b_{j_2}\rangle \geq t$, then

$$\begin{aligned}
\beta\langle a_i, b_{j_1}, c_{j_2}\rangle/\widetilde{d} &= 0.5 \cdot \beta\langle a_i, b_{j_1}, c_{j_2}\rangle/d \\
&\geq 0.25 \cdot \beta t/d,
\end{aligned}$$

where the 1st step follows from $2d = \widetilde{d}$ (see Eq. (2)). This means as desired that

$$\begin{aligned}
u_i &\geq \exp(0.25\beta t/d)/(4n^2\tau) - \widetilde{t} \\
&= 3\widetilde{t} - \widetilde{t} \\
&= 2\widetilde{t} \\
&= \widetilde{t}_0.
\end{aligned}$$

**Case 2.**

For a given $i \in [n]$, if for all $(j_1, j_2) \in [n] \times [n]$ we have $\langle a_i, b_{j_1}, c_{j_2}\rangle < t(1 - \epsilon)$, then

$$\beta\langle a_i, b_{j_1}, c_{j_2}\rangle/\widetilde{d} \leq 0.25\beta \cdot (1 - \epsilon)t/d.$$

Hence, as desired,

$$\begin{aligned}
u_i &< (n^2 \cdot \exp(0.25\beta(1 - \epsilon)t/d)))/(n^2\tau) + \widetilde{t} \\
&= \exp(0.25\beta t/d)/(4n^2\tau) \cdot (4n^2/\exp(0.25\beta\epsilon t/d)) + \widetilde{t} \\
&= 3\widetilde{t} \cdot (4n^2/\exp(0.25\beta\epsilon t/d)) + \widetilde{t} \qquad\qquad \text{by definition of } \widetilde{t}, \text{ see Eq. (8)} \\
&\leq 3\widetilde{t} \cdot \frac{1}{4} + \widetilde{t}
\end{aligned}$$

$$= 2\widetilde{t}$$
$$= \widetilde{t}_0.$$

Here, the 4th step follows because, by our choice of $\beta$ and $t$, we have

$$\begin{aligned}
\exp(0.25\beta\epsilon t/d) &= \exp((0.25\beta\epsilon C_0 \log n)/d) \\
&= \exp(0.25\beta\epsilon C_0/C) \\
&= \exp(10\log n) \\
&> 16n^2,
\end{aligned} \tag{9}$$

where we used that $t = C_0 \log n$ (by Lemma statement), that $d = C \log n$, that $\beta = B^3$ (Eq. (5)) and the choice of $B$ (Eq. (3)). $\qquad\square$

## B.2 MAIN HARDNESS RESULT

We can finally conclude our main lower bound.

**Theorem B.2** (Lower bound, formal version of Theorem 1.3). *Assuming* SETH, *for every $q > 0$, there are constants $C, C_a, C_b > 0$ such that: there is no algorithm running in time $O(n^{3-q})$ for the problem* AAttC$(n, d = C \log n, B = C_b \sqrt[3]{\log n}, \epsilon_a = n^{-C_a})$.

*Proof.* Follows from combining Theorem 4.5, Theorem 4.7, and Lemma B.1. $\qquad\square$

## C UPPER BOUND: RUNNING TIME

In Section C.1, we review the standard "matrix" attention computation problem. In Section C.2, we define the "tensor" attention computation problem. In Section C.3, we provide an efficient tool for implementing tensor related computations. In Section C.4, we provide several tools for rearranging tensor computations that we will use in our algorithm.

## C.1 CLASSICAL ATTENTION COMPUTATION

We first review the attention computation definition in Vaswani et al. (2017); Devlin et al. (2018); Radford et al. (2018; 2019); Brown et al. (2020); OpenAI (2023); Zandieh et al. (2023); Alman & Song (2023); Brand et al. (2023); Gao et al. (2023b;c;a),

**Definition C.1.** *Suppose there are three $n \times d$ size matrices $Q, K, V \in \mathbb{R}^{n \times d}$, our plan is to generate the $n \times d$ matrix* Att$(Q, K, V)$ *defined by*

$$\mathsf{Att}(\underbrace{Q}_{n\times d}, \underbrace{K}_{n\times d}, \underbrace{V}_{n\times d}) := \underbrace{\underbrace{D^{-1}}_{n\times n} \underbrace{A}_{n\times n} \underbrace{V}_{n\times d}}_{n\times d}$$

*where $A \in \mathbb{R}^{n \times n}$ and diagonal matrix $D \in \mathbb{R}^{n \times n}$ are defined as*

$$\underbrace{A}_{n\times n} := \exp(\underbrace{\underbrace{Q}_{n\times d} \underbrace{K^\top}_{d\times n}/d}_{n\times n}), \quad \text{and} \quad \underbrace{D}_{n\times n} := \mathrm{diag}(\underbrace{\underbrace{A}_{n\times n} \underbrace{\mathbf{1}_n}_{n\times 1}}_{n\times n})$$

## C.2 TENSOR ATTENTION COMPUTATION

Given two $n \times d$ matrices, there are two standard variants on their Kronecker product one may consider: The standard Kronecker product (denoted $\otimes$) is a new $n^2 \times d^2$ matrix, whereas the column-wise Kronecker product (denoted $\oslash$) is a new $n^2 \times d$ matrix. For more literature on tensor computations and their applications in learning algorithms, we refer the readers to Bhaskara et al. (2014); Song et al. (2019); Diao et al. (2018; 2019); Yang (2019; 2020a;b); Bhaskara et al. (2020); Ahle et al. (2020); Song et al. (2021a;b); Yang & Hu (2021); Song et al. (2023a;b;c); Deng et al. (2023).

Next, we generalize the matrix attention computation (in Alman & Song (2023)) into tensor attention computation as follows:

**Definition C.2** (TensorAtt). *Given matrices $Q, K_1, K_2 \in \mathbb{R}^{n \times d}$ and matrices $V_1, V_2 \in \mathbb{R}^{n \times d}$, the goal of tensor attention computation is to compute*

$$\underbrace{\mathsf{TensorAtt}(Q, K_1, K_2, V_1, V_2)}_{n \times d} := \underbrace{\underbrace{D^{-1}}_{n \times n} \underbrace{A}_{n \times n^2} \underbrace{V}_{n^2 \times d}}_{n \times d}$$

*where*

- $A \in \mathbb{R}^{n \times n^2}$ *is defined as* $A := \underbrace{Q}_{n \times d} \big( \underbrace{K_1}_{n \times d} \oslash \underbrace{K_2}_{n \times d} \big)^\top$

- $D \in \mathbb{R}^{n \times n}$ *is defined as* $D := \mathrm{diag}( \underbrace{A}_{n \times n^2} \cdot \underbrace{\mathbf{1}_{n^2}}_{n^2 \times 1} )$

- $V \in \mathbb{R}^{n^2 \times d}$ *is defined as* $V := \underbrace{V_1}_{n \times d} \oslash \underbrace{V_2}_{n \times d}$

## C.3 EFFICIENT COLUMN-WISE KRONECKER COMPUTATION

We prove an important tool which will be used in analyze the running time of our algorithm.

**Lemma C.3.** *If the following condition holds*

- *Let $\oslash$ be defined as Definition 2.4.*

- *Given $A_1 \in \mathbb{R}^{n \times d_1}$, $A_2 \in \mathbb{R}^{n \times d_1}$, we define $A := (A_1 \oslash A_2) \in \mathbb{R}^{n^2 \times d_1}$.*

- *Given $B_1 \in \mathbb{R}^{n \times d_2}$, $B_2 \in \mathbb{R}^{n \times d_2}$, we define $B := (B_1 \oslash B_2) \in \mathbb{R}^{n^2 \times d_2}$.*

- *We define $C \in \mathbb{R}^{d_1 \times d_2}$ as $C := A^\top B$*

- *We define $C_1 := A_1^\top B_1, C_2 := A_2^\top B_2$*

*Then, we have*

- *Part 1. $C_1 \circ C_2 = C$*

- *Part 2. Given as input $A_1, A_2, B_1, B_2$, we can compute $C$ in $\mathcal{T}_{\mathrm{mat}}(d_1, n, d_2)$ time.*

*Proof.* For each $i \in [n]$, let $a_{1,i}^\top$ denote the $i$-th row of $A_1 \in \mathbb{R}^{n \times d_1}$.

For each $i \in [n]$, let $a_{2,i}^\top$ denote the $i$-th row of $A_2 \in \mathbb{R}^{n \times d_1}$.

For each $i \in [n]$, let $b_{1,i}^\top$ denote the $i$-th row of $B_1$.

For each $i \in [n]$, let $b_{2,i}^\top$ denote the $i$-th row of $B_2$.

For each $i \in [d]$, let $A_{*,i} \in \mathbb{R}^{n^2}$ denote the $i$-th column of matrix $A \in \mathbb{R}^{n^2 \times d_1}$

Recall that $C_1 \in \mathbb{R}^{d_1 \times d_2}$ and $C_2 \in \mathbb{R}^{d_1 \times d_2}$,

$$C_1 := A_1^\top B_1, \quad C_2 := A_2^\top B_2$$

Thus, we see that

$$(C_1)_{k_1, k_2} = \sum_{i=1}^{n} a_{1,i,k_1} b_{1,i,k_2}$$

$$(C_2)_{k_1, k_2} = \sum_{j=1}^{n} a_{2,j,k_1} b_{2,j,k_2}$$

Then, we can write $C \in \mathbb{R}^{d_1 \times d_2}$ as

$$
\begin{aligned}
\underbrace{C}_{d_1 \times d_2} &= \underbrace{A^\top}_{d_1 \times n^2} \underbrace{B}_{n^2 \times d_2} \\
&= \sum_{i=1}^{n^2} \underbrace{A_{*,i}}_{d_1 \times 1} \underbrace{B_{*,i}^\top}_{1 \times d_2} \\
&= \sum_{i=1}^{n} \sum_{j=1}^{n} \underbrace{A_{*,i+(j-1)n}}_{d_1 \times 1} \underbrace{B_{*,i+(j-1)n}^\top}_{1 \times d_2} \\
&= \sum_{i=1}^{n} \sum_{j=1}^{n} \underbrace{(a_{1,i} \circ a_{2,j})}_{d_1 \times 1} \cdot \underbrace{(b_{1,i} \circ b_{2,j})^\top}_{1 \times d_2}
\end{aligned} \tag{10}
$$

where the first step follows from definition of $C \in \mathbb{R}^{d \times d}$, the second step follows from the matrix can written as the summation of $n^2$ rank-1 matrices, the third step follows from changing the index, the forth step follows from $\underbrace{A_{*,i+(j-1)n}}_{d_1 \times 1} = \underbrace{a_{1,i}}_{d_1 \times 1} \circ \underbrace{a_{2,j}}_{d_1 \times 1}$.

From the above, we can calculate that the entry of $C$ in location $k_1, k_2$ is

$$
\begin{aligned}
C_{k_1,k_2} &= \sum_{i=1}^{n} \sum_{j=1}^{n} (a_{1,i} \circ a_{2,j})_{k_1} \cdot (b_{1,i} \circ b_{2,j})_{k_2}^\top \\
&= \sum_{i=1}^{n} \sum_{j=1}^{n} a_{1,i,k_1} a_{2,j,k_1} b_{1,i,k_2} b_{2,j,k_2} \\
&= \left( \sum_{i=1}^{n} a_{1,i,k_1} b_{1,i,k_2} \right) \cdot \left( \sum_{j=1}^{n} a_{2,j,k_1} b_{2,j,k_2} \right) \\
&= (C_1)_{k_1,k_2} \cdot (C_2)_{k_1,k_2}
\end{aligned}
$$

where the first step follows from Eq. (10), the second step follows from simple algebra, the third step follows from separating the summation over $i$ and the summation over $j$, and the last step follows from definition of matrices $C_1$ and $C_2$.

Thus, we can conclude

$$
C = C_1 \circ C_2.
$$

The algorithm will first compute $C_1$ and $C_2$, whic takes $\mathcal{T}_{\mathrm{mat}}(d_1, n, d_2)$ time. Then it calculates $C_1 \circ C_2$, which takes $O(d_1 d_2)$ time. $\qquad \square$

### C.4 SIMPLE EQUIVALENT TOOLS FOR TENSOR NOTATIONS

We define a standard tensor notation, for example see Song et al. (2019).

**Definition C.4** $((\cdot, \cdot, \cdot)$ tensor operator). *Given a tensor $T \in \mathbb{R}^{n_1 \times n_2 \times n_3}$, let $X \in \mathbb{R}^{n_1 \times d_1}$, $Y \in \mathbb{R}^{n_2 \times d_2}$, $Z \in \mathbb{R}^{n_3 \times d_3}$.*

*We define $T(X, Y, Z) \in \mathbb{R}^{d_1 \times d_2 \times d_3}$ as follows*

$$
T(X, Y, Z)_{i,j,l} = \sum_{a=1}^{n_1} \sum_{b=1}^{n_2} \sum_{c=1}^{n_3} T_{a,b,c} X_{a,i} Y_{b,j} Z_{c,l}, \quad \forall a \in [d_1], b \in [d_2], c \in [d_3].
$$

Next, we present several equivalence results for tensors.

**Lemma C.5.** *If the following conditions hold*

- *Let $\oslash$ be defined as Definition 2.4.*

- *Let $\odot$ be defined as Definition 2.2.*

- *Let $(\cdot, \cdot, \cdot)$ operator be defined as Definition C.4.*

- *Let $\circ$ be defined as Definition 2.1.*

- *Let $Q, K_1, K_2, V_1, V_2 \in \mathbb{R}^{n \times d}$*

- *Let $A \in \mathbb{R}^{n \times n^2}$ be $Q(K_1 \oslash K_2)^\top$.*

- *Let $\mathsf{A} \in \mathbb{R}^{n \times n \times n}$ be $Q \odot K_1 \odot K_2$.*

*Then, we have*

- **Part 1.** $A_{i, j_1 + (j_2 - 1)n} = \mathsf{A}_{i, j_1, j_2}$ *for $i \in [n], j_1 \in [n], j_2 \in [n]$ (This means $\mathsf{A}$ can be viewed as the tensor version of $A$)*

- **Part 2.** $A \mathbf{1}_{n^2} = \mathsf{A}(I, \mathbf{1}_n, \mathbf{1}_n) = Q \odot (\mathbf{1}_n^\top K_1) \odot (\mathbf{1}_n^\top K_2)$

- **Part 3.** $A(V_1 \oslash V_2) = Q(K_1 \oslash K_2)^\top (V_1 \oslash V_2) = Q((K_1^\top V_1) \circ (K_2^\top V_2))$

*Proof.* **Proof of Part 1.**

Directly follows from definition of $A$ and $\mathsf{A}$.

**Proof of Part 2.**

Follows from tensor notations in Definition 2.2 and Definition C.4.

**Proof of Part 3.**

Directly follows from applying Part 1 of Lemma C.3 here. $\qquad\square$

## D  UPPER BOUND: ERROR ANALYSIS

In Section D.1, we provide the definition of approximate tensor attention computation. In Section D.2, we state a polynomial approximation tool from previous work. In Section D.3, we show a bound on the entries of the attention matrix. In Section D.4, we provide a low-rank decomposition for the tensor version of the attention matrix. Finally, in Section D.5, we compute the error propagation from $A$ to $D$, then in Section D.6, we analyze the error propagation from $A$ and $D$ to the attention matrix.

### D.1  APPROXIMATE TENSOR ATTENTION COMPUTATION

**Definition D.1** (A tensor generalization of standard attention computation, restatement of Definition 1.2)**.** *Let $\epsilon_a > 0$, $B > 0$ be parameters. Given five matrices $Q, K_1, K_2, V_1, V_2 \in \mathbb{R}^{n \times d}$ such that*

- $\|Q\|_\infty \leq B, \|K_1\|_\infty \leq B, \|K_2\|_\infty \leq B, \|V_1\|_\infty \leq B, \|V_2\|_\infty \leq B$

*Our goal is to find a matrix $T \in \mathbb{R}^{n \times d}$ which can entry-wisely approximate $D^{-1}AV$, in particular, it means the following $\ell_\infty$ norm guarantee,*

$$\|T - D^{-1}AV\|_\infty \leq \epsilon_a$$

*Here,*

- $A := \exp(Q(K_1 \oslash K_2)^\top) \in \mathbb{R}^{n \times n^2}$ *(We remark that we can also view matrix $A$ as the flattening of an $n \times n \times n$ tensor)*

- $V := V_1 \oslash V_2 \in \mathbb{R}^{n^2 \times d}$

- $D = \operatorname{diag}(A \mathbf{1}_{n^2}) \in \mathbb{R}^{n \times n}$ *is an $n \times n$ size positive diagonal matrix.*

Notice that the straightforward algorithm for this problem will spend at least $\Omega(n^3)$ time to write the matrix $A$ (we can also think of $A$ as an tensor that has size $n \times n \times n$).

## D.2 An Error Control Tool From Previous Work

We state a tool from previous work.

**Corollary D.2** (Corollary 2.2 in Alman & Song (2023)). *Suppose the following conditions hold*

- *Let $B > 1$.*

- *Let $\epsilon \in (0, 0.1)$.*

- *Let $g := \Theta(\max\{\frac{\log(1/\epsilon)}{\log(\log(1/\epsilon)/B)}, B\})$.*

*There is a polynomial $P : \mathbb{R} \to \mathbb{R}$ of degree-$g$ such that for all $x \in [-B, B]$, we have*

$$(1 - \epsilon) \cdot \exp(x) < P(x) < (1 + \epsilon) \cdot \exp(x).$$

## D.3 Tensor $Q \odot K_1 \odot K_2$ Has Bounded Entries

**Lemma D.3** (Bounded entry). *Suppose the following conditions hold*

- *Suppose $B \geq 1$*

- *Assume matrices $Q, K_1, K_2 \in \mathbb{R}^{n \times d}$ have $\|Q\|_\infty \leq B$, $\|K_1\|_\infty \leq B$, $\|K_2\|_\infty \leq B$.*

- *Let $\odot$ operation be defined as Definition 2.2.*

*Then, we have*

$$\|Q \odot K_1 \odot K_2/d\|_\infty \leq B^3.$$

*Proof.* For every index triple $(i, j_1, j_2) \in [n] \times [n] \times [n]$, we are able to prove

$$
\begin{aligned}
|(QK^\top)_{i,j_1,j_2}| = |\sum_{l=1}^{d} Q_{i,l}(K_1)_{j_1,l}(K_2)_{j_2,l}| \\
\leq d \cdot \|Q\|_\infty \cdot \|K_1\|_\infty \cdot \|K_2\|_\infty \\
\leq d \cdot B^3,
\end{aligned}
$$

where the 2nd step is because triangle inequality, the 3rd step is using $\ell_\infty$ norm bound on $Q, K_1, K_2$.

Now, we complete the proofs. $\square$

## D.4 Tensor Low-Rank Approximation

In the following definition, we view the $n \times n^2$ size matrix as an $n \times n \times n$ size attention matrix.

**Definition D.4.** *Assume the following parameters regime,*

- *We use $r \geq 1$ to denote a positive integer.*

- *We use $\epsilon \in (0, 0.1)$ to represent an accuracy parameter.*

- *Suppose there is a 3rd order tensor $\mathsf{A} \in \mathbb{R}_{\geq 0}^{n \times n \times n}$*

*We say 3rd order tensor $\widetilde{\mathsf{A}} \in \mathbb{R}_{\geq 0}^{n \times n \times n}$ is an $(\epsilon, r)$-approximation of tensor $\mathsf{A}$ if*

- *$\widetilde{\mathsf{A}} = U_1 \odot U_2 \odot U_3$ for some matrices $U_1, U_2, U_3 \in \mathbb{R}^{n \times r}$ (i.e., $\widetilde{\mathsf{A}}$ has rank at most $r$), and*

- *$|\widetilde{\mathsf{A}}_{i,j_1,j_2} - \mathsf{A}_{i,j_1,j_2}| \leq \epsilon \cdot \mathsf{A}_{i,j_1,j_2}$ for all $(i, j_1, j_2) \in [n] \times [n] \times [n]$.*

### D.5 FROM $A$ TO $D$

In this section and the next, we generalize the proof of Alman & Song (2023) for error propagation from the matrix setting to the tensor setting. The proofs are nearly identical.

**Lemma D.5.** *Let $A \in \mathbb{R}^{n \times n^2}$ be a matrix with positive entries, and $\epsilon_A \in (0, 0.1)$ be any parameter. Let $\widetilde{A} \in \mathbb{R}^{n \times n^2}$ be an approximation to $A$, meaning for all $(i, l) \in [n] \times [n^2]$, we have*

$$|\widetilde{A}_{i,l} - A_{i,l}| \leq \epsilon_A \cdot A_{i,l}.$$

*We consider two diagonal matrices $D, \widetilde{D} \in \mathbb{R}^{n \times n}$ which can be formally written as $D := \mathrm{diag}(A\mathbf{1}_{n^2})$ and $\widetilde{D} := \mathrm{diag}(\widetilde{A}\mathbf{1}_{n^2})$.*

*Then, for every index $i \in [n]$, the following bound holds*

$$|\widetilde{D}_{i,i} - D_{i,i}| \leq \epsilon_A \cdot D_{i,i}.$$

*Proof.* We calculate that

$$
\begin{aligned}
|\widetilde{D}_{i,i} - D_{i,i}| &= |\sum_{l=1}^{n^2} \widetilde{A}_{i,l} - \sum_{j=1}^{n^2} A_{i,l}| \\
&\leq \sum_{l=1}^{n^2} |\widetilde{A}_{i,l} - A_{i,l}| \\
&\leq \sum_{l=1}^{n^2} \epsilon_A A_{i,l} \\
&= \epsilon_A \cdot D_{i,i}.
\end{aligned}
$$

where the second step follows from triangle inequality.

This completes the proof. $\square$

### D.6 FROM $A, D$ TO TENSOR ATTENTION

The goal of this section is to prove Lemma D.6.

**Lemma D.6.** *Suppose the following conditions are true*

- *Let $\epsilon_A, \epsilon_D \in (0, 0.1)$*

- *Let $B > 1$ be a bounded parameter,*

- *We use $V = (V_1 \oslash V_2) \in \mathbb{R}^{n^2 \times d}$ to represent a matrix with $\|V\|_\infty \leq B^2$.*

- *Let $A \in \mathbb{R}_{>0}^{n \times n^2}$ be a positive matrix,*

- *and let $\widetilde{A} \in \mathbb{R}^{n \times n^2}$ be a matrix such that, for every tuple $(i, l) \in [n] \times [n^2]$ we have*

$$|\widetilde{A}_{i,l} - A_{i,l}| \leq \epsilon_A \cdot A_{i,l}.$$

- *Suppose $D, \widetilde{D} \in \mathbb{R}^{n \times n}$ are diagonal matrices with positive diagonal entries, and such that for every index $i \in [n]$, we have*

$$|\widetilde{D}_{i,i} - D_{i,i}| \leq \epsilon_D \cdot D_{i,i}.$$

*Then, we have*

$$\|\widetilde{D}^{-1}\widetilde{A}V - D^{-1}AV\|_\infty \leq (\epsilon_A + \epsilon_D) \cdot B^2.$$

*Proof.* By the triangle inequality, we know

$$\|\widetilde{D}^{-1}\widetilde{A}V - D^{-1}AV\|_\infty \leq \|\widetilde{D}^{-1}\widetilde{A}V - D^{-1}\widetilde{A}V\|_\infty + \|D^{-1}\widetilde{A}V - D^{-1}AV\|_\infty. \quad (11)$$

We bound each of these two terms to get our desired result.

First of all, for every index pair $(i,j) \in [n] \times [d]$,

$$
\begin{aligned}
|(\widetilde{D}^{-1}\widetilde{A}V - D^{-1}\widetilde{A}V)_{i,j}| &= |\sum_{l=1}^{n^2}(\widetilde{D}_{i,i}^{-1} - D_{i,i}^{-1}) \cdot \widetilde{A}_{i,l} \cdot V_{l,j}| \\
&\leq \sum_{l=1}^{n^2} |(\widetilde{D}_{i,i}^{-1} - D_{i,i}^{-1}) \cdot \widetilde{A}_{i,l}| \cdot \|V\|_\infty \\
&= \sum_{l=1}^{n^2} |\frac{D_{i,i} - \widetilde{D}_{i,i}}{D_{i,i}\widetilde{D}_{i,i}} \widetilde{A}_{i,l}| \cdot \|V\|_\infty \\
&\leq \epsilon_D \cdot \sum_{l=1}^{n^2} |\widetilde{D}_{i,i}^{-1}\widetilde{A}_{i,l}| \cdot \|V\|_\infty \\
&= \epsilon_D \cdot |\sum_{l=1}^{n^2} \widetilde{D}_{i,i}^{-1}\widetilde{A}_{i,l}| \cdot \|V\|_\infty \\
&= \epsilon_D \cdot \|V\|_\infty \\
&\leq \epsilon_D \cdot B^2.
\end{aligned}
\quad (12)
$$

Here the 2nd step uses the triangle inequality, the 4th step follows from the assumption that $|(D_{i,i} - \widetilde{D}_{i,i})/D_{i,i}| \leq \epsilon_D$, the 5th step follows because $\widetilde{D}_i^{-1}$ and $\widetilde{A}_{i,l}$ are positive numbers, and the final step follows by $\ell_\infty$ norm of $V$ is bounded by $B^2$.

Second, for every $(i,j) \in [n] \times [d]$,

$$
\begin{aligned}
|(D^{-1}\widetilde{A}V - D^{-1}AV)_{i,j}| &= |\sum_{l=1}^{n^2} D_{i,i}^{-1}(\widetilde{A}_{i,l} - A_{i,l}) \cdot V_{l,j}| \\
&\leq \sum_{l=1}^{n^2} |D_{i,i}^{-1}| \cdot |(\widetilde{A}_{i,l} - A_{i,l})| \cdot \|V\|_\infty \\
&= \sum_{l=1}^{n^2} D_{i,i}^{-1} \cdot |(\widetilde{A}_{i,l} - A_{i,l})| \cdot \|V\|_\infty \\
&\leq \sum_{l=1}^{n^2} D_{i,i}^{-1} \cdot \epsilon_A A_{i,l} \cdot B^2 \\
&= \epsilon_A \cdot B^2.
\end{aligned}
\quad (13)
$$

Here, again, the 2nd step uses the triangle inequality, the 3rd step follows because $D_{i,i}^{-1}$ is positive, the 4th step follows from the assumption that $|\widetilde{A}_{i,l} - A_{i,l}| \leq \epsilon_A \cdot A_{i,l}$ and the final step follows by definition of $D_{i,i}$.

The Lemma conclusion then becomes true by substituting Eq. (12) and Eq. (13) into Eq. (11). $\qquad\square$

## E  UPPER BOUND: PUTTING IT ALL TOGETHER

In Section E.1, we provide a low-rank decomposition for approximating the original attention tensor. In Section E.2, we calculate the running time of constructing that low-rank decomposition. In Section E.3, we put everything together, and prove our main upper bound theorem.

### E.1 Decomposing into $U_1, U_2$ and $U_3$

The goal of this section is to prove Lemma E.1.

**Lemma E.1.** *If the following conditions hold*

- *We use $M := X \odot Y \odot Z \in \mathbb{R}^{n \times n \times n}$ to represent a tensor that is constructed by $X, Y, Z \in \mathbb{R}^{n \times d}$.*

- *Let $P(x)$ denote a degree-$g$ single-variable polynomial. We apply $P(M)$ entry-wisely, i.e, $P(M)_{i,j,l} = P(M_{i,j,l})$.*

- *Let $r$ be rank parameter that is $r := \binom{3(g+d)}{3g}$.*

*There is an algorithm running in time $O(nrg)$ which, given as input $X, Y, Z$, constructs matrices $U_1, U_2, U_3 \in \mathbb{R}^{n \times r}$ such that $P(M) = U_1 \odot U_2 \odot U_3$.*

*Proof.* Expand $P$ as a sum of monomials as

$$P(x) = \sum_{i=0}^{d} c_i \cdot x^i.$$

Consider the function $\mathsf{K} : \mathbb{R}^d \times \mathbb{R}^d \times \mathbb{R}^d \to \mathbb{R}$ defined by, for $u, v, w \in \mathbb{R}^d$,

$$\mathsf{K}(u, v, w) := P(\langle u, v, w \rangle).$$

We define set $V$ and provide names for variables in set $V$ in the following sense,

$$V := \{u_1, \cdots, u_d, v_1, \cdots, v_d, w_1, \cdots, w_d\}.$$

Thus, function $\mathsf{K}$ can be viewed a degree-$3g$ polynomial in the $3d$ entries in $V$ of the vectors $u, v, w$.

We count the number of its monomials.

We define set $\mathcal{F}$ as

$$\mathcal{F} := \left\{ f : V \to \{0, 1, 2, \cdots, 3g\} \mid \sum_{v \in V} f(v) \leq 3g \right\}.$$

Let us count the size of set $\mathcal{F}$

$$|\mathcal{F}| = \binom{3d + 3g}{3g}.$$

There exists coefficients $\{c_t\}_{t \in \mathcal{F}} \in \mathbb{R}$ such that

$$\mathsf{K}(u, v, w) = \sum_{t \in \mathcal{F}} c_t \cdot \prod_{\beta \in V} v^{t(\beta)}.$$

We define partitions of $V$:

$$V_u := \{u_1, \cdots, u_d\}, \quad V_v := \{v_1, \cdots, v_d\}, \quad V_w := \{w_1, \cdots, w_d\}.$$

We define $\phi_u : \mathbb{R}^d \to \mathbb{R}^{|\mathcal{F}|}$ by, for any $t \in \mathcal{F}$,

$$\phi_u(u_1, \cdots, u_d)_t = c_t \cdot \prod_{u_i \in V_u} u_i^{t(u_i)}.$$

Similarly, we define $\phi_v : \mathbb{R}^d \to \mathbb{R}^{|\mathcal{F}|}$ by, for any $t \in \mathcal{F}$,

$$\phi_v(v_1, \cdots, v_d)_t = \prod_{v_i \in V_v} v_i^{t(v_i)}.$$

and we define $\phi_w : \mathbb{R}^d \to \mathbb{R}^{|\mathcal{F}|}$ by, for any $t \in \mathcal{F}$,

$$\phi_w(w_1, \cdots, w_d)_t = \prod_{w_i \in V_w} w_i^{t(w_i)}.$$

Here, we can view $\mathsf{K}$ function as

$$\mathsf{K}(u, v, w) = \langle \phi_u(u), \phi_v(v), \phi_w(w) \rangle.$$

For every index $i \in [n]$, suppose $X_i \in \mathbb{R}^d$ is the $i$-th row of $X$, assume $Y_i \in \mathbb{R}^d$ is the $i$-th row of $Y$, and let $Z_i \in \mathbb{R}^d$ denote the $i$-th row of $Z$.

Therefore, we should construct three matrices $U_1, U_2$ and $U_3$ as the following way, for each $i \in [n]$

- the $i$-th row of the matrix $U_1 \in \mathbb{R}^{n \times |\mathcal{F}|}$ is the vector $\phi_u(x_i)$,

- the $i$-th row of the matrix $U_2 \in \mathbb{R}^{n \times |\mathcal{F}|}$ is the vector $\phi_v(y_i)$,

- the $i$-th row of the matrix $U_3 \in \mathbb{R}^{n \times |\mathcal{F}|}$ is the vector $\phi_w(z_i)$.

These $n \times r$ matrices can be constructed in time $O(nrg)$ in the straightforward way, since each entry depends on $g$ variables. □

---

**Algorithm 1** Our Polynomial Method Tensor Attention Algorithm

1: **procedure** POLYTENSORATTENTION($Q \in \mathbb{R}^{n \times d}, K_1 \in \mathbb{R}^{n \times d}, K_2 \in \mathbb{R}^{n \times d}, V_1 \in \mathbb{R}^{n \times d}, V_2 \in \mathbb{R}^{n \times d}, n \in \mathbb{N}_+, d \in \mathbb{N}_+, B > 0, \epsilon \in (0, 0.1)$)        ▷ Theorem 1.4
2:       ▷ $n$ can be viewed as the length of the sentence
3:       ▷ $d$ can be viewed as the feature of dimension
4:       ▷ $\epsilon$ is the accuracy output
5:       ▷ $\max\{\|Q\|_\infty, \|K_1\|_\infty, \|K_2\|_\infty, \|V_1\|_\infty, \|V_2\|_\infty\} \le B$
6:     $g \leftarrow O(\max\{\frac{\log(1/\epsilon)}{\log(\log(1/\epsilon)/B^3)}, B^3\})$
7:     $r \leftarrow \binom{3(g+d)}{3d}$
8:     /*Step 1*/
9:     Construct $U_1, U_2, U_3 \in \mathbb{R}^{n \times r}$ via Lemma E.2       ▷ $O(nrg)$ time
10:    /*Step 2*/
11:    $\widetilde{w} \leftarrow \underbrace{U_1}_{n \times r} \cdot (\underbrace{(U_2 \oslash U_3)^\top}_{r \times n^2} \underbrace{(\mathbf{1}_n \oslash \mathbf{1}_n)}_{n^2 \times 1})$       ▷ $O(nr)$ time
12:    /*Step 3*/
13:    $\widetilde{D}^{-1} = \text{diag}(\widetilde{w}^{-1})$       ▷ $O(n)$ time
14:    /*Step 4*/
15:    Compute $(U_2 \oslash U_3)^\top (V_1 \oslash V_2) \in \mathbb{R}^{r \times d}$       ▷ Takes $\mathcal{T}_{\text{mat}}(r, n, d)$ time
16:    /*Step 5*/
17:    Compute $U_1 \cdot ((U_2 \oslash U_3)^\top (V_1 \oslash V_2))$       ▷ $\mathcal{T}_{\text{mat}}(n, r, d)$ time
18:    /*Step 6*/
19:    $T \leftarrow \widetilde{D}^{-1} \cdot (U_1 \cdot ((U_2 \oslash U_3)^\top (V_1 \oslash V_2)))$       ▷ $O(nd)$ time
20:    **return** $T$       ▷ $T \in \mathbb{R}^{n \times d}$
21: **end procedure**

---

### E.2 TIME FOR CONSTRUCTING $U_1, U_2, U_3$

**Lemma E.2.** *Suppose five matrices* $Q, K_1, K_2, V_1, V_2 \in \mathbb{R}^{n \times d}$ *satisfy*

- $\|Q\|_\infty \le B$,

- $\|K_1\|_\infty \le B, \|K_2\|_\infty \le B,$

- $\|V_1\|_\infty \le B,$ *and* $\|V_2\|_\infty \le B.$

*We define tensor* $\mathsf{A} := \exp(Q \odot K_1 \odot K_2/d) \in \mathbb{R}^{n \times n \times n}$.

*For bounded number $B > 0$ and accuracy parameter $\epsilon \in (0, 1)$, there are positive integers $g$ and $r$*

- *the condition for $g$:*

$$g = O\Big( \max \Big\{ \frac{\log(1/\epsilon)}{\log(\log(1/\epsilon)/B^3)}, B^3 \Big\} \Big),$$

- *the condition for $r$:*

$$r \le \binom{3(g+d)}{3g}$$

*such that:*

*There is a third order tensor $\widetilde{\mathsf{A}} \in \mathbb{R}^{n \times n \times n}$ that*

- *$\widetilde{\mathsf{A}}$ is an $(\epsilon, r)$-approximation (Definition D.4) of $\mathsf{A} \in \mathbb{R}^{n \times n \times n}$*

- *Let $U_1$, $U_2$ and $U_3 \in \mathbb{R}^{n \times r}$ be the matrices defining $\widetilde{\mathsf{A}}$, i.e., $\widetilde{\mathsf{A}} = U_1 \odot U_2 \odot U_3$*

- *it takes $O(nr)$ time to construct $U_1, U_2$ and $U_3$.*

*Proof.* We define $M := Q \odot K_1 \odot K_2/d \in \mathbb{R}^{n \times n \times n}$. Using Lemma D.3, we can show that
$$\|M\|_\infty \le B^3.$$

Recall that the definition of $(\epsilon, r)$-approximation can be found in Definition D.4.

Next, we will apply Corollary D.2 (with replacing $B$ by $B^3$), there is a degree-$g$ polynomial $P : \mathbb{R} \to \mathbb{R}$ such that the tensor $\widetilde{\mathsf{A}} = P(M)$ is an $(\epsilon, r)$-approximation to tensor $\mathsf{A}$. Here we apply $P$ to $M$ entrywisely.

We can then compute $U_1, U_2$, and $U_3$ using Lemma E.1, which gives the bound
$$r \le \binom{3(g+d)}{3g}.$$

Therefore, we finish the proof. $\qquad\square$

### E.3 MAIN ALGORITHMIC RESULT

We present our main algorithmic result as follows:

**Theorem E.3** (Upper bound, formal version of Theorem 1.4)**.** *There is an algorithm (Algorithm 1) that solves* $\mathsf{ATAttC}(n, d = O(\log n), B = o(\sqrt[3]{\log n}), \epsilon_a = 1/\operatorname{poly}(n))$ *in time* $n^{1+o(1)}$.

*Proof.* **Proof of Running Time.**

- Using Lemma E.2, we know that Step 1 (in Algorithm 1) can be implemented in $O(nrg)$ time

- Using Lemma C.3, we know that Step 2 (in Algorithm 1) can be implemented in $O(nr)$ time

- Step 3 can implemented in $O(n)$ in a straightforward way.

- To compute Step 4 efficiently, we need to use Lemma C.3 again.

- Computing Step 5 is just standard matrix multiplication

- Step 6 is just rescaling the $n \times d$ matrix

**Proof of Correctness.**

We combine Corollary D.2, Lemma D.5, Lemma D.6, and simple algebra. $\qquad\square$

