# OpenReview forum: "How to Capture Higher-order Correlations? Generalizing Matrix Softmax Attention to Kronecker Computation"
_ICLR.cc/2024/Conference — ICLR 2024 spotlight_

### Official Review · Reviewer_7Xxi · 2023-10-21

**Soundness:** 3 good
**Presentation:** 4 excellent
**Contribution:** 3 good
**Rating:** 8
**Confidence:** 3

**Summary:**

This paper solves the transformer attention scheme over triplets under some mild complexity theoretical assumptions.

**Strengths:**

Originality:
This work is original to the best of my knowledge.

Quality:
Quality is high.

Clarity:
Writing is clear.

Significance:
The results of this paper are important due to their connections to LLMs and other AI applications.

**Weaknesses:**

None.

**Questions:**

Page 2:
Could you please explain the column-wise Kronecker product of V_1 and V_2 here as well?

Page 3:
Please elaborate on SETH.
I know it is standard, but perhaps too strong? :)

Page 4:
Please provide some intuition for Definition 1.5.

Page 5:
I do not understand "Approximating A."

Page 6:
Could you please sketch an short example for the reduction from GapMaxIP to ATAttc?

Page 8:
Could you please add some discussion about Theorem 4.7?

Page 9:
Line 12 from bottom:
Why is there such a \mu?

---

> ### Author Response · Authors · 2023-11-20
> **Reply to Reviewer 7Xxi**
>
> Thank you to the reviewer for the feedback and detailed suggestions.
>
> Q: Could you please explain the column-wise Kronecker product of V_1 and V_2 here as well?
>
> A: Yes, thanks.
>
> Q: Please elaborate on SETH. I know it is standard, but perhaps too strong? :)
>
> A: SETH roughly says that one cannot improve too much on our current best algorithms for solving CNF-SAT. This is one of the most well-studied algorithms problems, yet improvements have been rare and very small. Although the word “strong” appears in its name, the majority of complexity theorists believe it is true, and much of the area of fine-grained complexity theory is built off of it.
>
>
>
> Q: I do not understand "Approximating A."
>
> A: The matrix $\widetilde{A}$ is an ell_infinity approximation of A.
>
> Q: Could you please sketch an short example for the reduction from GapMaxIP to ATAttc?
>
> A: The reduction is depicted at the bottom of page 22; we will refer to it above.
>
> Q: Could you please add some discussion about Theorem 4.7?
>
> A: Yes, this is the result we discuss in section 3.2 (“Hardness of Gap-MaxIP”).
>
> Q: Line 12 from bottom: Why is there such a $\mu$?
>
> A: Sorry, there is a typo here. If the three are _not_ orthogonal then there is no such $\mu$, and if they are orthogonal, then there is such a $\mu$. The message $\mu$ is the one Merlin would send to get the players to accept that the vectors are orthogonal.

---

### Official Review · Reviewer_FrM6 · 2023-10-30

**Soundness:** 3 good
**Presentation:** 3 good
**Contribution:** 3 good
**Rating:** 8
**Confidence:** 3

**Summary:**

The paper studies the computational aspects of a so-called "third-order tensor attention variant" recently defined by Sanford et al (2023). While standard attention captures pairwise interactions between tokens, the third-order tensor variant was suggested in the context of capturing triplet interactions.

The current paper shows that if the input entries are bounded, and if it suffices to compute each entry of the attention output approximately rather than exactly, then there is an almost-linear time algorithm for computing this operation. The bound on the entry magnitude is asymptotically (conditionally) tight, as the paper also shows that without this bound, computing this attention output in time significantly better than the trivial n^3 time computation is SETH-hard, in the complexity theoretic sense. Both the algorithm and the hardness result are generalizations (in terms of both the results themselves, and the techniques used to prove them) of a recent work of Alman and Song (2023) that proved them for standard attention. The results also extend to tensors of higher orders (than 3).

**Strengths:**

The paper is generally well-written and the mathematical content seems interesting.

**Weaknesses:**

The paper is purely theoretical and seems quite removed from application. It is entirely about a form of tensor attention that has been suggested as a bit of an afterthought in a recent work (Sanford 2023), that deemed it likely impractical and anyway did not implement it. Thus it is not about an architecture that is actually used or presently considered usable. This raises the question of what is the import of showing the existence of an almost-linear time approximation algorithm for it, and whether this algorithm makes sense as part of a neural network (I believe the paper does not touch on this point). A negative outlook on this paper would be that the interesting results and the conceptual message in this line of research were already given in Alman and Song (2023), and the extension to higher-order tensors in this manuscript might be an elegant intellectual and mathematical exercise, albeit without much consequence to the ML community. Nonetheless, since the content does seem elegant, and there is always the issue of benefit-of-doubt about whether a piece of theoretical research would have implications down the road, I prefer to vote for acceptance.

**Questions:**

I'd be interested to hear from the authors what do they consider to be the importance and implication of their algorithm, and whether they deem it implementable within a neural network architecture?

---

> ### Author Response · Authors · 2023-11-20
> **Reply to Reviewer FrM6**
>
> Thank you to the reviewer for the feedback and detailed suggestions.
>
> Q: I'd be interested to hear from the authors what do they consider to be the importance and implication of their algorithm, and whether they deem it implementable within a neural network architecture?
>
>
> A: One of the biggest takeaways for us is that tensor attention can actually, perhaps unintuitively, be computed in near-linear time. The prior work by Sanford et al. proposed tensor attention as a solution to the weaknesses of matrix attention which they studied, but they remarked that its unclear how to compute it quickly, and thus stopped considering it. We hope our result that it actually could be computed quickly will spark more work on determining both the theoretical and practical use of this generalization.
>
> We also find it intriguing that the bound on the entries which one needs to fast tensor attention decreases with the dimension of the tensor: for dimension k tensors, it grows like $(\log n)^{1/k}$. This may lead to a tradeoff in future work: one would pick the largest $k$ for which their entries are small enough, and thus get the most expressive version of attention which can still be computed quickly.
>
> Whether it is implementable within a neural network architecture will evidently require more engineering work beyond our theoretical paper, but we are optimistic that it is possible, as the polynomial method is not ultimately too complicated.

---

> > ### Comment · Reviewer_FrM6 · 2023-11-22
> >
> > I thank the authors for their answers.

---

### Official Review · Reviewer_dthX · 2023-11-02

**Soundness:** 3 good
**Presentation:** 2 fair
**Contribution:** 3 good
**Rating:** 8
**Confidence:** 3

**Summary:**

This work explores the computational complexity of generalized matrix attention scheme, which is used to capture high-order correlation among the input sequence.
To capture the correlation among triplets of tokens in an input sequence of length $n$, the generalized attention scheme outputs an attention matrix by computing the column-wise Kronecker products based on one query matrix, two key matrices and two value matrices.
In such a case, this work shows if the max entries of the query, key and value matrices are bounded by $o(\sqrt[3]{\log n})$, one can compute an approximation to the generalized attention matrix in almost linear time. On the other hand, if the max entries of the input matrices are at least $\Omega(\sqrt[3]{\log n}$), one cannot hope to efficiently approximate  the attention matrix in less than cubic time, assuming SETH. The latter hardness result is shown by reducing from the Gap Max IP problem, whose hardness is then shown through a combination and generalization of previous techniques.  Furthermore, the work shows the techniques developed above can be extended to characterize the gap in computational complexity in $k$-th order generalization of the attention scheme.

**Strengths:**

- This work considers an interesting problem of computing generalized matrix attention scheme. Since the generalized schemes involves computing the Kronecker products between a set of matrices, this is apparently a computationally expensive operation. It is hence natural to explore the computational complexity of this problem.

- Both the upper bound and the lower bound results (especially the latter one) presented in the work are interesting.

- A high-level summarization and intuition behind the techniques used to derive the upper and the lower bound helps the reader.

**Weaknesses:**

The presentation needs to be improved.

- In Section 3.2 hardness, “3-Max IP” and “Gap-MaxIP” seem to refer to the same problem. It is confusing to give two names to the same problem.

- It might be clearer to present the upper bound (UB) and the lower bound (LB) in two separate sections, instead of giving an overview of the UB + LB, and then elaborate on the LB.

- Is it possible to give a few sentences of description of the mysterious $U_1, U_2, U_3$ matrices and how to compute them in Section 3.1?

- In Section 4 “hardness” which elaborates on key steps in showing the LB, presenting Hypothesis 4.1, Definition 4.2, Conjecture 4.3 and Theorem 4.4, all of which are from prior works, in the main paper do not help much on understanding and appreciating the novelty / challenges addressed in extending and generalizing current proof techniques to show the LB on computational complexity for approximating the generalized attention matrix. Some of them can indeed be moved to the Appendix. It would be better to give more intuition on the (technical) difference between the three-party and four-party communication protocol that computes set disjointness, how algebraic geometry code is applied in extending the proof from three-party to four-party communication and how the new protocol is used in showing the LB on computation time of Gap Max-IP.

- Minor issue: the last paragraph in page 5 states “showing that the same line of attack”? “attack” here means “techniques”, I assume?

**Questions:**

I am not a complexity expert. I have no comments on the proof techniques presented in this work. However, I do have a few questions for the authors.

- In Definition 1.2, why is approximating the higher order attention matrix in the $\ell_{\infty}$ norm considered a good metric to evaluate the approximation of a matrix that captures higher order correlation? Why not the other norms?

- In Section 3.1, does “$\widetilde{D} \approx D$” mean $\tilde{D}$ and $D$ close in the $\ell_{\infty}$ norm? (and so does “$\widetilde{A} \approx A$”?)

- In Section 3.1, why can $\widetilde{A}(V_1 / V_2)$ be computed in $O(n^{1+o(1)})$ time, while $\widetilde{D}$ needs to be computed in $O(nd)$ time?

- Why does the construction of the algorithm in Section 3.1 fail when there are large entries $\Omega(\sqrt[3]{\log n})$ in the input matrices?

- In Section 4, what is the major challenge of extending the three-party communication protocol to a four-party communication protocol in Section 4.2? Why does one need to use the algebraic geometry code?

- In Section 4, where does $B = O(\sqrt[3]{\log n})$ pop up in the LB proof?

---

> ### Author Response · Authors · 2023-11-20
> **Reply to Reviewer dthX**
>
> Thank you to the reviewer for the feedback and detailed suggestions.
>
> Q: In Section 3.2 hardness, “3-Max IP” and “Gap-MaxIP” seem to refer to the same problem. It is confusing to give two names to the same problem.
>
> A: Yes, our new problem “Gap-MaxIP” is a promise version of the more common “3-Max IP” problem from the literature. We’ve changed the heading of the paragraph to no longer say “3-MaxIP”, and discuss it in the text instead.
>
> Q: It might be clearer to present the upper bound (UB) and the lower bound (LB) in two separate sections, instead of giving an overview of the UB + LB, and then elaborate on the LB.
>
> A: Thank you. We arranged the paper as we did to fit in the page limit, but will try a clearer arrangement for the camera ready version.
>
> Q: Is it possible to give a few sentences of description of the mysterious $U_1, U_2, U_3$ matrices and how to compute them in Section 3.1?
>
> A: Yes we can elaborate on this. Just as showing a matrix is low-rank is equivalent to showing it is the product of two rectangular matrices, showing that a tensor is low-rank is equivalent to showing that it is the “product” of three rectangular matrices; $U_1, U_2, U_3$ are those three matrices.
>
> Q: In Section 4 “hardness” which elaborates on key steps in showing the LB, presenting Hypothesis 4.1, Definition 4.2, Conjecture 4.3 and Theorem 4.4, all of which are from prior works, ...... the proof from three-party to four-party communication and how the new protocol is used in showing the LB on computation time of Gap Max-IP.
>
> A: We tried to focus in the technique overview on the novel ideas here that don’t already appear in prior work on these communication protocols and fine-grained hardness results. One of the main challenges here was to determine the correct definitions and generalizations so that techniques like AG codes from prior work continue to apply, but the problem can still be reduced to tensor attention. See below for more details.
>
> Q: Minor issue: the last paragraph in page 5 states “showing that the same line of attack”? “attack” here means “techniques”, I assume?
>
> A: Yes, by “line of attack” we mean “sequence of techniques”.
>
>
> Q: In Definition 1.2, why is approximating the higher order attention matrix in the $\ell_{\infty}$ norm considered a good metric to evaluate the approximation of a matrix that captures higher order correlation? Why not the other norms?
>
> A: The $ell_\infty$ norm can be translated into other reasonable norms, our dependence on eps in the running time is only $\log(1 / \epsilon)$. Thus, for instance, if you pay $10 \log(n/ \epsilon)$ in the running time, you will get $\epsilon / n^{10}$ error. Then, if you have $\ell_\infty$ norm only $\epsilon / n^{10}$, then you also have very small matrix spectral norm and matrix  Frobenius norm bounded by $( \epsilon / n^{10} )  * n^2$.
>
> Q: In Section 3.1, does “$\widetilde{D} \approx D$” mean $\widetilde{D}$ and $D$ close in the  norm? (and so does $\widetilde{A}$ and $A$?)
>
> A: Yes.
>
> Q: In Section 3.1, why \widetilde{A} (V_1 V2) can be computed in $n^{1+o(1)}$  time, while $\widetilde{D}$ needs to be computed in $O(nd)$  time?
>
> A: Here $d = O(\log n)$, so $O(nd) = O(n \log n)$ is faster than $n^{1 + o(1)}$.
>
> Q: Why does the construction of the algorithm in Section 3.1 fail when there are large entries  in the input matrices?
>
> A: The bound on B is needed to have a low-degree polynomial approximation of the exponential function. The larger B is, the higher degree one needs, and eventually it gets too large and the approach no longer gives a nontrivial algorithm.
>
> Q: In Section 4, what is the major challenge of extending the three-party communication protocol to a four-party communication protocol in Section 4.2? Why does one need to use the algebraic geometry code?
>
> A: The challenging aspect is to determine what the correct problem is that the four players should be solving. Prior work ultimately showed hardness of an “approximate closest pair” problem, which does not easily generalize to yield hardness for our tensor setting. Indeed, natural options like “approximate closest triple” aren’t easy to reduce to attention. Once the correct problem definitions are in place, the actual protocol is not so difficult given prior work. AG codes are important here to save log factors compared to simpler constructions like Reed-Solomon codes. Saving the log factor is important since it ultimately appears in the exponent of the running time of our reduction in section E.
>
> Q: In Section 4, where does $B = O(\sqrt[3]{\log n})$  pop up in the LB proof?
>
> A: The bound B on the entries in attention arises in section E in the appendix, where we reduce from Gap-MaxIP to attention and complete the hardness proof. Section 4 proves hardness of Gap-MaxIP, and doesn’t yet involve the entry bound.

---

> > ### Comment · Reviewer_dthX · 2023-11-20
> > **Response to authors' comments**
> >
> > Thank you very much for the clarification. I would like to raise the score from 6 to8.

---

### Meta-Review · Area_Chair_VJ6c · 2023-12-07

**Metareview:**

This paper proposes an attention matrix that is able to capture higher-order correlations between words in a sentence. Computational issues of approximately computing the proposed matrix with high accuracy and computational efficiency is discussed.

**Justification For Why Not Higher Score:**

Some experimental results to demonstrate the validity of the developed theory would be beneficial.

**Justification For Why Not Lower Score:**

All 3 reviewers agree that the work addresses an important problem and provided mathematically rigorous results.

---

### Decision · Program_Chairs · 2024-01-16

Accept (spotlight)